# Chronic glucose-dependent insulinotropic polypeptide receptor (GIPR) agonism desensitizes adipocyte GIPR activity mimicking functional GIPR antagonism

Elizabeth A. Killion [1], Michelle Chen[1], James R. Falsey[2], Glenn Sivits [1], Todd Hager[3], Larissa Atangan[1], Joan Helmering[1], Jae Lee[1], Hongyan Li[3], Bin Wu[2], Yuan Cheng[2], Murielle M. Véniant[1] & David J. Lloyd [1✉]

Antagonism or agonism of the glucose-dependent insulinotropic polypeptide (GIP) receptor (GIPR) prevents weight gain and leads to dramatic weight loss in combination with glucagon-like peptide-1 receptor agonists in preclinical models. Based on the genetic evidence supporting GIPR antagonism, we previously developed a mouse anti-murine GIPR antibody (muGIPR-Ab) that protected diet-induced obese (DIO) mice against body weight gain and improved multiple metabolic parameters. This work reconciles the similar preclinical body weight effects of GIPR antagonists and agonists in vivo, and here we show that chronic GIPR agonism desensitizes GIPR activity in primary adipocytes, both differentiated in vitro and adipose tissue in vivo, and functions like a GIPR antagonist. Additionally, GIPR activity in adipocytes is partially responsible for muGIPR-Ab to prevent weight gain in DIO mice, demonstrating a role of adipocyte GIPR in the regulation of adiposity in vivo.

[1] Amgen Research, Department of Cardiometabolic Disorders, Amgen Inc., One Amgen Center Dr, Thousand Oaks, CA 91320, USA. [2] Amgen Research, Department of Selection and Modality Engineering, Amgen Inc., One Amgen Center Dr, Thousand Oaks, CA 91320, USA. [3] Amgen Research, Department of Translational Safety & Bioanalytical Sciences, Amgen Inc., One Amgen Center Dr, Thousand Oaks, CA 91320, USA. ✉email: dlloyd@amgen.com

Glucose-dependent insulinotropic polypeptide (GIP) and glucagon-like peptide-1 (GLP-1) are gut-derived incretin hormones, known to stimulate insulin secretion for glycemic control, and in the case of GLP-1, also recognized to promote satiety, which has led to the development of a GLP-1 receptor (GLP-1R) agonist (GLP-1RA) as an obesity therapy[1]. More recently, therapies targeting the glucose-dependent insulinotropic polypeptide (GIP) receptor (GIPR) have shown efficacy preclinically in preventing weight gain[2–5].

The GIPR locus has been identified in genome-wide association studies to be associated with obesity and body-mass index (BMI)[6] highlighting its importance as a regulator of adiposity in humans. Alleles have been identified that both increase[7] and, more importantly, decrease BMI[8], presenting support for potential GIPR-directed therapies as weight loss agents. Furthermore, in some studies, the lower BMI alleles have been associated with either reduced expression[6], signaling[9,10], or incretin function[2,11,12]. In alignment with the human genetic evidence, mouse gene deletion studies of GIP, GIPR, or ablation of GIP-secreting K cells all demonstrate protection from diet-induced obesity (DIO)[13–16].

Based on the human and mouse genetic evidence supporting GIPR antagonism[6], we previously developed anti-GIPR antagonistic antibodies as a potential therapeutic strategy for the treatment of obesity. A mouse anti-murine anti-GIPR antibody (muGIPR-Ab) protected DIO mice against body weight gain, improved multiple metabolic parameters, and was associated with reduced food intake and resting respiratory exchange ratio[2]. Interestingly, preclinical studies utilizing GIPR agonists[3–5] display a similar response to muGIPR-Ab both alone and in combination with GLP-1RAs[2]. Moreover, the dual GIP/GLP-1 analog tirzepatide has demonstrated enhanced weight loss both preclinically and clinically beyond GLP-RAs alone[3,17], intensifying the scientific debate surrounding the use of GIPR agonists or antagonists for the treatment of obesity[6].

The purpose of this work is to reconcile the similar preclinical body weight effects of GIPR antagonists and agonists in vivo, and here we show that a long-acting-(LA)-GIPR agonist (LA-Agonist) desensitizes GIPR activity in primary adipocytes, both differentiated in vitro and adipose tissue in vivo, and functions like a GIPR antagonist. Additionally, we establish that GIPR activity in adipocytes is partially responsible for the ability of muGIPR-Ab to prevent weight gain in DIO mice, demonstrating a role of adipocyte GIPR in the regulation of adiposity in vivo.

## Results

**LA-Agonist has the same effect on body weight as muGIPR-Ab.** To compare the effect of a GIPR agonist head-to-head with the GIPR antagonist muGIPR-Ab alone and in combination with GLP-1RA liraglutide, we developed a tool molecule with high potency and improved pharmacokinetic (PK) parameters that combines a modified GIP peptide with an antibody against a non-mammalian target to ensure maximal activation of the GIPR. First, we tested our long-acting-(LA)-GIPR Agonist (LA-Agonist) in vitro and determined its activity in cells overexpressing mouse GIPR compared to GIP (Fig. 1a) and determined its selectivity for GIPR over GLP-1 receptor (GLP-1R) and glucagon receptor (Supplemental Fig. 1a–c). Using a pharmacodynamic (PD) assay with a GIP analog [D-Ala$^2$]-GIP (DA-GIP) as a control, DIO mice were injected intraperitoneal (IP) with glucose and saline, glucose and DA-GIP, or glucose and the LA-Agonist in a dose response to determine the PD effect. The LA-Agonist was more potent at lowering blood glucose (Fig. 1b) and increasing insulin secretion (Fig. 1c) at 50 and 150 nmol/kg compared to DA-GIP (50 nmol/kg). We then established an

exposure-PD response relationship for blood concentration of the LA-Agonist vs. the area under the curve for both glucose and insulin (Fig. 1d, e). The LA-Agonist half maximal inhibitory and effective concentration (IC$_{50}$ and EC$_{50}$) was 328 nM for glucose and 212 nM for insulin, respectively. Utilizing a single-dose PK study, the terminal half-life and bioavailability for the intact LA-Agonist following IP injection were determined to be 71.3 h and ≈100%, respectively (Fig. 1f and Supplemental Table 1). In conjunction with the PK-PD response, a 2-compartment PK model was used to simulate exposure in a multiple dose efficacy study. The dose and dosing frequency were evaluated to provide target coverage greater than the IC$_{50}$ for glucose response (328 nM). To ensure that we conservatively maintained maximal efficacy, we selected a daily dose of the LA-Agonist ten times higher than a dose needed to cover the IC$_{50}$ of 328 nM (37.5 mg/kg/day) at trough concentrations to compare against muGIPR-Ab. We previously reported that the muGIPR-Ab dose-dependently inhibited GIP-stimulated cAMP in vitro in the same assay reported here in Fig. 1a with IC$_{50}$ = 89.6 nM, and in vivo, the maximum effect in the acute PD assay was achieved with muGIPR-Ab (25 mg/kg), which correlated with a mean serum concentration of 2250 nM, and allowed us to determine that muGIPR-Ab dosed 25 mg/kg every six days was sufficient to provide maximal target coverage[2]. Since 37.5 mg/kg of the LA-Agonist (equivalent to 239.5 nmol/kg LA-Agonist) is a supra-pharmacological dose, we confirmed that both DA-GIP and the peptide portion of the LA-Agonist both at 250 nmol/kg robustly stimulated insulin secretion in DIO mice, which was absent in DIO mice with pancreatic Gipr β-cell knockout (Gipr$^{\beta Cell−/−}$; mice previously described[2]) (Supplemental Fig. 1d, e), confirming that high concentrations of GIPR agonists in vivo do not activate GLP-1R mediated insulin secretion.

During chronic treatment in DIO mice fed 60% kcal from fat diet for 12 weeks, the LA-Agonist and muGIPR-Ab both demonstrated protection against body weight gain, while the combination of LA-Agonist and the GLP-1RA liraglutide and the combination of muGIPR-Ab and liraglutide groups both equally lost a greater amount of weight than the combined effect of either monotherapy alone (Fig. 2a). By measuring food intake over time, all treatment groups, except muGIPR-Ab alone, showed a significant reduction in food intake during treatment days 1–3 compared to vehicle (Fig. 2b). During treatment days 7–9, only groups receiving liraglutide alone or in combination showed reduced food intake, which returned to the same level as vehicle by the end of the study. Notably, during days 13–15, food intake for both the LA-Agonist in combination with liraglutide and muGIPR-Ab in combination with liraglutide groups was still decreased whereas it was not different in any of the monotherapy groups.

On day 18, only mice that received liraglutide alone or in combination had reduced fat mass, and LA-Agonist treated mice did not differ from muGIPR-Ab treated mice alone or in combination with liraglutide (Fig. 2c). At the end of the study, both the LA-Agonist in combination with liraglutide and the muGIPR-Ab in combination with liraglutide groups had reduced subcutaneous WAT (scWAT) weights (Fig. 2d). All treatment groups, except the LA-Agonist alone, had reduced epididymal WAT (eWAT) weights at necropsy, and the LA-Agonist did not differ from muGIPR-Ab alone or in combination with liraglutide (Fig. 2e). Only mice treated with liraglutide alone or in combination had reduced liver weights (Fig. 2f). Reflective of reduced fat mass, all treatment groups had reduced plasma leptin and resistin (Fig. 2g, h). Minimal differences from vehicle were observed in plasma adiponectin and only mice treated with muGIPR-Ab in combination with liraglutide had increased adiponectin (Fig. 2i).

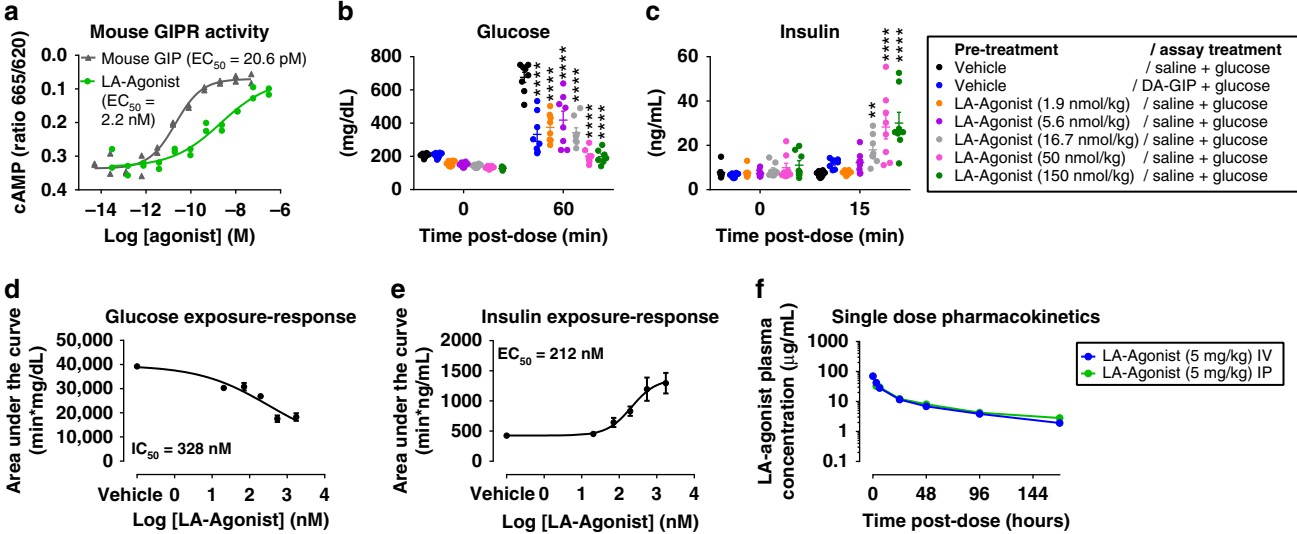

**Fig. 1 Generation of a long-acting GIPR agonist (LA-Agonist) for chronic in vivo studies. a** GIPR agonist activity of LA-Agonist and mouse GIP in cells overexpressing mouse GIPR determined by measuring cAMP. $n = 2$ wells/treatment. **b, c** Pharmacodynamic (PD) assay for LA-Agonist-stimulated insulin secretion. DIO mice were injected with saline or LA-Agonist, fasted for 4 h, injected with glucose and saline or glucose and DA-GIP as indicated. **b** Blood glucose and **c** plasma insulin were measured over time. $n = 8$ mice/group; two-way repeated measures ANOVA with Dunnett's multiple comparisons test, $**p = 0.0048$ and $****p \leq 0.0001$ vs. vehicle/saline. Data are presented as mean values ± SEM. **d, e** LA-Agonist exposure from samples taken at 4-h post dose of LA-Agonist in the same PD assay presented in **b, c** were correlated to individual response for **d** glucose and **e** insulin, based on 0–60 min AUC calculated from **b** and **c**, respectively. $n = 8$ mice/treatment. Data are presented as mean values ± SEM. **f** Single-dose pharmacokinetic (PK) study of LA-Agonist was evaluated in CD-1 mice both by intravenous (IV) and intaperitoneal (IP) injections. Plasma was collected over time and used to measure intact LA-Agonist. $n = 3$ mice/timepoint. Data are presented as mean values ± SEM.

To determine the effect of GIPR agonism or antagonism on GIPR incretin activity after 21 days, we conducted a PD assay by injecting glucose and DA-GIP into all mice, and, as expected, the muGIPR-Ab treated mice did not have increased insulin concentrations after DA-GIP stimulation whereas the LA-Agonist treated mice increased insulin concentrations to a similar extent as vehicle-treated animals, indicating intact incretin activity in LA-Agonist treated animals (Fig. 2j). All other treatment groups had significantly reduced DA-GIP-stimulated insulin concentrations (Fig. 2j). The LA-Agonist alone and both combination groups had significantly reduced blood glucose after 80 min post-dose (Fig. 2k).

**LA-Agonist and muGIPR-Ab have similar plasma metabolomics.** To determine the metabolic alterations of GIPR agonism compared to antagonism independent of body weight differences, i.e., the LA-Agonist alone and muGIPR-Ab alone both had a modest effect preventing body weight gain while the combination with liraglutide yielded dramatic body weight loss, we utilized untargeted metabolomics analysis of plasma samples from mice chronically treated with vehicle, LA-Agonist, or muGIPR-Ab collected 80 min after the glucose + DA-GIP challenge (shown in Fig. 2). All plasma metabolomics data is available in Supplemental Data 1. Compared to vehicle, plasma from mice treated with muGIPR-Ab had $n = 254$ metabolites out of 671 detected metabolites significantly different from vehicle ($p \leq 0.05$ and $q \leq 0.2$; Fig. 3a) and plasma from mice treated with LA-Agonist had $n = 257$ different metabolites out of 671 detected metabolites significantly different from vehicle ($p \leq 0.05$ and $q \leq 0.2$; Fig. 3b). Notably, there were not any significantly different metabolites detected between muGIPR-Ab compared to LA-Agonist treated mice ($p \leq 0.05$ and $q \leq 0.2$; Fig. 3c). High level views of the metabolome revealed that the plasma metabolic profiles from the LA-Agonist and muGIPR-Ab treated mice were similar utilizing principal component analysis (PCA; Fig. 3d). Plasma samples

from mice treated with LA-Agonist or muGIPR-Ab clustered similarly on the PCA plot and were distinct from plasma samples from vehicle-treated mice (Fig. 3d), indicating limited differences between GIPR agonism and antagonism signaling in vivo. Using pathway enrichment analysis, the pathways found to be enriched in muGIPR-Ab compared to vehicle were also found to be enriched in LA-Agonist compared to vehicle with the highest enrichment for both treatments including amino sugar metabolism; fatty acid metabolism (acyl choline); and fructose, mannose, and galactose metabolism (Fig. 3e).

Since the plasma metabolomes were similar between muGIPR-Ab and LA-Agonist treated mice, we compared metabolites with known relationships to GIPR activity, notably cyclic adenosine monophosphate (cAMP)[2], corticosterone[18], and glycerol[19]. All three of these metabolites have been shown to be increased by GIP stimulation[2,18,19] and therefore were expected to be increased in LA-Agonist treated mice and decreased in muGIPR-Ab-treated mice. However, all three of these metabolites were reduced after DA-GIP stimulation in both LA-Agonist and muGIPR-Ab treated mice compared to vehicle (Fig. 3f–h).

To confirm these reported relationships from the literature, we stimulated naïve DIO mice with DA-GIP after an overnight fast, and measured plasma corticosterone (Fig. 3i) and glycerol (Fig. 3j) concentrations. We observed a dose-dependent increase in corticosterone secretion and determined DA-GIP $EC_{90} = 80$ nmol/kg. To confirm the GIPR-specific response of these metabolites, we pre-treated naïve DIO mice with muGIPR-Ab or vehicle and repeated the assay using 80 nmol/kg DA-GIP. Treatment with muGIPR-Ab significantly inhibited both DA-GIP-stimulated corticosterone secretion (Fig. 3k) and glycerol secretion (Fig. 3l), confirming that acute GIPR stimulation should increase both metabolites in vivo and muGIPR-Ab antagonizes this effect.

To confirm the effect of GIP to stimulate cAMP production, we determined a GIP dose-dependent response in mouse primary adipocytes differentiated in vitro for cAMP concentrations

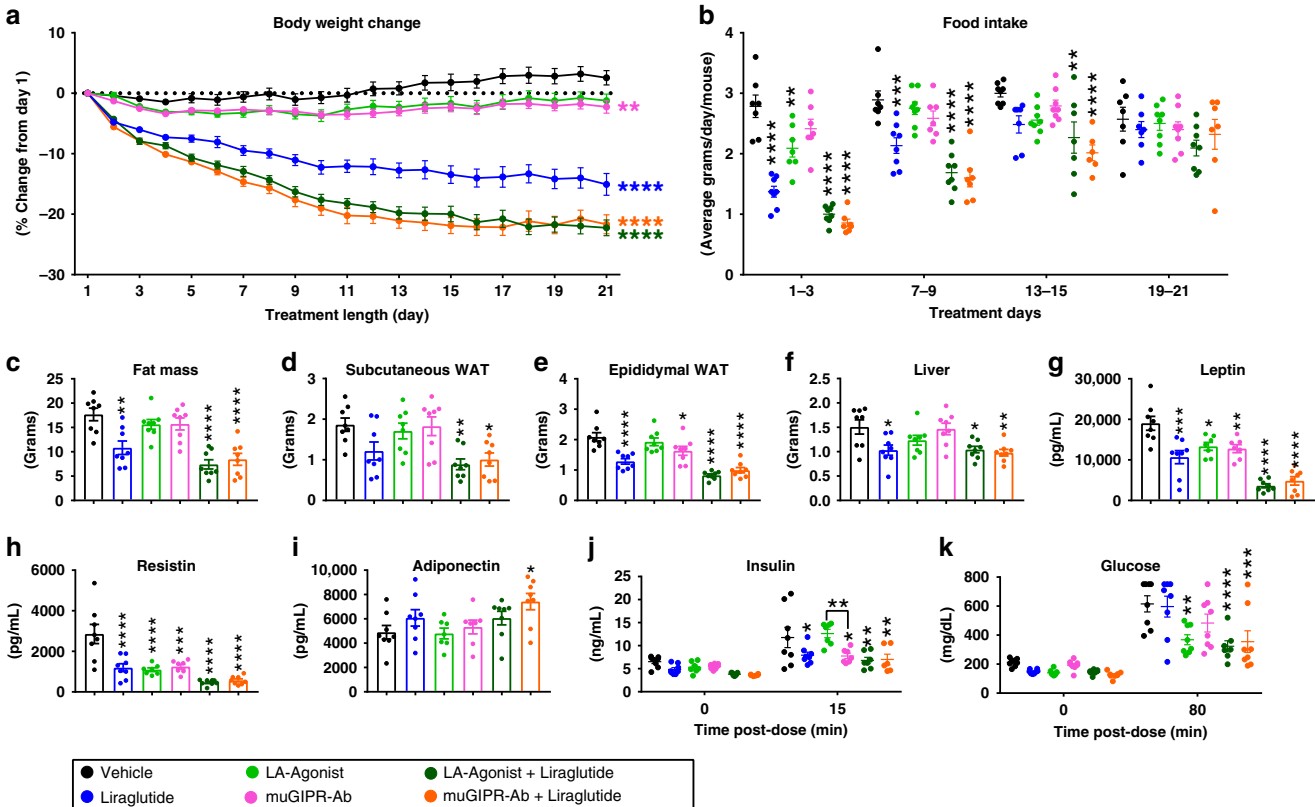

**Fig. 2 LA-Agonist has the same effect on body weight as muGIPR-Ab. a–k** DIO mice dosed with vehicle, liraglutide, LA-Agonist, muGIPR-Ab, LA-Agonist + liraglutide, or muGIPR-Ab + liraglutide for 21 days. **a** Body weight change from day 1. On day 21, Vehicle vs. Liraglutide ****$p \leq 0.0001$, Vehicle vs. LA-Agonist $p = 0.077$, Vehicle vs. muGIPR-Ab **$p = 0.0082$, Vehicle vs. Liraglutide + LA-Agonist ****$p \leq 0.0001$, and Vehicle vs. Liraglutide + muGIPR-Ab ****$p \leq 0.0001$. **b** 3-day average food intake measured over time. On days 1–3, **$p = 0.0089$ and ****$p \leq 0.0001$; on days 7–9, ***$p = 0.0020$ and ****$p \leq 0.0001$; and on days 13–15, **$p = 0.0031$ and ****$p \leq 0.0001$. **c** Total fat mass measured by MRI on day 18. **$p = 0.0032$ and ****$p \leq 0.0001$. Tissue weights collected at necropsy **d** scWAT, **e** eWAT, and **f** liver weights. **d** **$p = 0.0044$ and *$p = 0.016$, **e** *$p = 0.040$ and ****$p \leq 0.0001$; and **f** Vehicle vs. Liraglutide *$p = 0.017$, Vehicle vs. LA-Agonist + Liraglutide *$p = 0.021$, and **$p = 0.0059$. Four-hour fasted plasma adipokines (**g**) leptin, (**h**) resistin, and (**i**) adiponectin collected on day 21. **g** *$p = 0.018$, **$p = 0.0071$, ***$p = 0.0001$, and ****$p \leq 0.0001$; **h** ***$p = 0.0001$ and ****$p \leq 0.0001$; and **i** *$p = 0.027$. **j** Plasma insulin and **k** blood glucose concentrations were measured before and after IP injections of DA-GIP and glucose. **j** At 15 min, Vehicle vs. Liraglutide *$p = 0.026$, Vehicle vs. muGIPR-Ab *$p = 0.024$, Vehicle vs. LA-Agonist + Liraglutide **$p = 0.0022$, Vehicle vs. muGIPR-Ab + Liraglutide **$p = 0.0064$, and LA-Agonist vs. muGIPR-Ab **$p = 0.0043$. **k** At 80 min, Vehicle vs. LA-Agonist ***$p = 0.0009$, Vehicle vs. LA-Agonist + Liraglutide ***$p = 0.0001$, and Vehicle vs. muGIPR-Ab + Liraglutide ***$p = 0.0004$. **a–k** $n = 8$ mice/group; **a**, **j**, **k** two-way repeated measures ANOVA with Tukey's multiple comparisons test; **b** two-way ANOVA with Tukey's multiple comparisons test; **c–i** one-way ANOVA with Sidak's multiple comparisons test. Data are presented as mean values ± SEM.

(EC$_{50}$ = 3.8 nM), and muGIPR-Ab dose-dependently inhibited this effect at submaximal concentrations of GIP (Fig. 3m, n). In adipocytes, high concentrations of cAMP leads to lipolysis of stored triglyceride released as glycerol into the bloodstream[20]. We then determined a GIP dose-dependent response on adipocyte glycerol release from primary mouse adipocytes differentiated in vitro (EC$_{50}$ = 13 nM), and muGIPR-Ab dose-dependently inhibited this effect at submaximal concentrations of GIP (Fig. 3o, p). Overall, previous reports[2,19] and our in vivo (Fig. 3h, j, l) and in vitro (Fig. 3o, p) data demonstrate that GIP stimulates GIPR-dependent increases in cAMP and lipolysis, and the chronic treatment of muGIPR-Ab in vivo antagonizes this effect. Of note, chronic treatment with the LA-Agonist did not increase in vivo cAMP, corticosterone, or glycerol as might be expected as seen with acute treatment with DA-GIP, but in fact reduced plasma concentrations. Taken together, reductions in metabolites known to be increased by GIP stimulation, including cAMP (Fig. 3f), corticosterone (Fig. 3g), and glycerol (Fig. 3h), suggests functional GIPR activity inhibition in adrenal glands and adipocytes following chronic LA-Agonist treatment, reflecting similar levels of metabolites seen in muGIPR-Ab treated mice. It is important

to note that there is no evidence in the literature that plasma cAMP reflects the action of a single Gs coupled G-protein coupled receptors (GPCRs), but the plasma cAMP data in Fig. 3f is unequivocally lower then controls and is similar for chronic GIPR agonism and antagonism.

**GIP stimulates adipocyte fatty acid uptake.** While GIP is known to stimulate lipolysis in the absence of insulin[19], physiological GIP concentrations likely do not act to promote lipolysis in vivo since the presence of high GIP will also stimulate insulin secretion, and insulin is a potent inhibitor of GIP-stimulated lipolysis[21]. Similar to insulin, GIP has also been reported to stimulate fatty acid (FA) uptake and re-esterification in rat adipose tissue[22,23] and in mouse primary adipocytes ex vivo[19,24]. Therefore, we confirmed that GIP acutely stimulated FA uptake into primary mouse adipocytes differentiated in vitro (Fig. 4a) and muGIPR-Ab inhibited this effect in a dose-dependent manner (Fig. 4b). We translated this effect in vivo by establishing a FA uptake assay in metabolic tissues of DIO mice using $^{14}$C-oleic acid in olive oil. Overnight treatment of muGIPR-Ab significantly inhibited DA-GIP's incretin effect (Fig. 4c). DA-GIP significantly

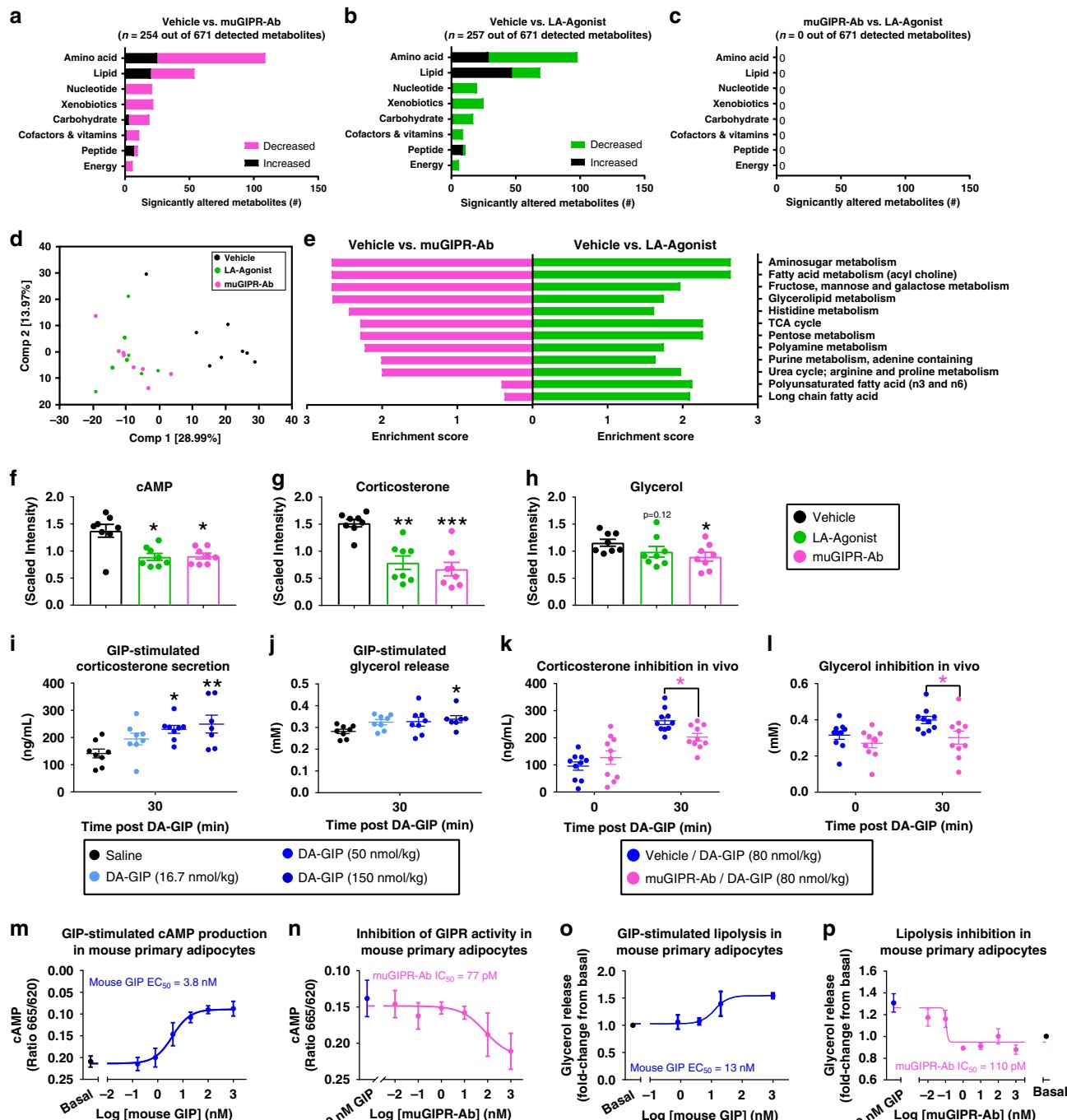

**Fig. 3 Chronic LA-Agonist treatment has similar plasma metabolomics as muGIPR-Ab treatment, associated with agonist-induced receptor desensitization. a–e** Global metabolomics analysis of plasma samples for vehicle, LA-Agonist, and muGIPR-Ab treated mice (presented in Fig. 2a–k). Significantly different number of metabolites between **a** vehicle vs. muGIPR-Ab, **b** vehicle vs. LA-Agonist, and **c** muGIPR-Ab vs. LA-Agonist. $n = 8$ mice/ group; Welch's two-sample $t$-test $p \leq 0.05$ and Storey method for correcting multiple comparisons $q$-value $\leq 0.2$. **d** Principal component analysis (PCA) of the plasma metabolites from global metabolomics analysis. **d** Significantly enriched pathways between vehicle vs. muGIPR-Ab and vehicle vs. LA-Agonist treatments. **f–h** Plasma metabolites **f** cAMP, **g** corticosterone, and **h** glycerol. Data are scaled such that the median value measured across all samples was set to 1.0 ± SEM, $n = 8$ mice/group; Welch's two-sample $t$-test, **f** Vehicle vs. LA-Agonist *$p = 0.0101$ and Vehicle vs. muGIPR-Ab *$p = 0.0121$; **g** **$p = 0.0023$, and ***$p = 0.000865$; **h** *$p = 0.029$. **i, j** DIO mice were injected with saline or DA-GIP, and blood was collected 30 mins post-dose to measure plasma metabolites **i** corticosterone and **j** glycerol. $n = 8$ mice/group; one-way ANOVA with Sidak's multiple comparisons test, **i** *$p = 0.017$ and **$p = 0.0047$, **j** *$p = 0.043$. Data are presented as mean values ± SEM. **k, l** DIO mice were pre-treated with vehicle or muGIPR-Ab 24-h before injection with saline or DA-GIP and plasma metabolites **k** corticosterone and **l** glycerol were measured over time. $n = 10$ mice/group; two-way repeated measures ANOVA with Sidak's multiple comparisons test, **k** *$p = 0.043$, **l** *$p = 0.029$. Data are presented as mean values ± SEM. **m, n** cAMP concentration in mouse primary adipocytes (**m**) treated with GIP or (**n**) pre-treated with muGIPR-Ab overnight then treated with 10 nM mouse GIP. Data represents means ± SEM of three mice with three wells/treatment/mouse. **o, p** GIP-stimulated lipolysis in primary differentiated mouse pre-adipocytes **o** treated with mouse GIP compared to basal or **p** after pre-treatment with muGIPR-Ab overnight then treated with 10 nM mouse GIP. Data represent means ± SEM of 3–6 mice with three wells/treatment/mouse.

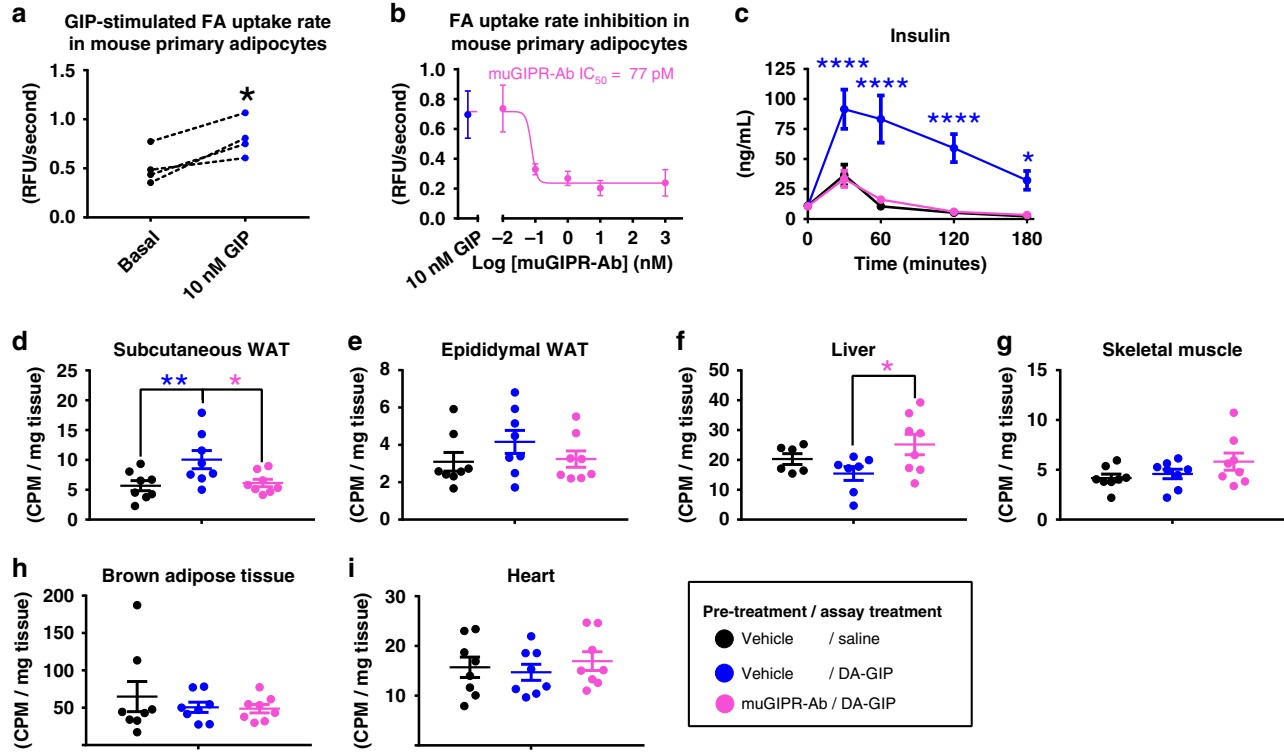

**Fig. 4 GIP stimulates fatty acid uptake in primary adipocytes in vitro and in adipose tissue in vivo and is inhibited by muGIPR-Ab. a** FA uptake rate in primary differentiated mouse pre-adipocytes treated with GIP for 30 mins then FA uptake rate measured. Data represent mean ± SEM of adipocytes from four mice, three wells/treatment/mouse; Student's paired *t* test, *$p = 0.023$. **b** FA uptake rate in primary differentiated mouse pre-adipocytes pre-treated with muGIPR-Ab or vehicle overnight then treatment with 10 nM GIP for 30 mins and FA uptake rate measured. Data represent mean ± SEM of adipocytes from three mice, three wells/treatment/mouse. **c–i** DIO mice were pre-treated with vehicle or muGIPR-Ab (25 mg/kg) for 24 h then injected with saline or DA-GIP and simultaneously oral gavaged with $^{14}$C-oleic acid in olive oil to assess FA uptake. **c** Plasma insulin measured over time, *$p = 0.038$ and ****$p \leq 0.0001$. Radioactivity uptake into metabolically relevant tissues was measured at necropsy 180 min post-dose in **d** scWAT, **e** eWAT, **f** liver, **g** skeletal muscle, **h** brown adipose tissue, and **i** heart. **d** Vehicle vs. DA-GIP *$p = 0.023$ and Vehicle vs. muGIPR-Ab/DA-GIP *$p = 0.043$, **f** *$p = 0.050$. $n = 8$ mice/ group; **c** two-way repeated measures ANOVA with Tukey's test for multiple comparisons (**c**) or (**d–i**) one-way ANOVA with Tukey's test for multiple comparisons test, *$p \leq 0.05$ and ****$p \leq 0.0001$. Data are presented as mean values ± SEM.

promoted FA uptake into scWAT (Fig. 4d) without an effect in eWAT (Fig. 4e), and this effect was blocked by muGIPR-Ab. Interestingly, mice treated with muGIPR-Ab had increased FA uptake into the liver compared to DA-GIP treatment alone (Fig. 4f). No alterations in FA uptake in skeletal muscle, brown adipose tissue, or heart were observed (Fig. 4g–i).

Since DA-GIP treatment in combination with olive oil by oral gavage leads to insulin secretion, the FA uptake observed in Fig. 4d could be a result of elevated insulin concentrations. Therefore, we utilized *Gipr* β-cell knockout mice (*Gipr*$^{βCell−/−}$) that do not secrete insulin in response to GIP stimulation in order to tease out the effect of GIP-stimulated FA uptake independently of DA-GIP-stimulated insulin secretion. In *Gipr* floxed control mice (*Gipr*$^{fl/fl}$), GIP significantly stimulated insulin secretion, and this effect was absent in muGIPR-Ab treated *Gipr*$^{fl/fl}$ mice and *Gipr*$^{βCell−/−}$ mice with either treatment (Supplemental Fig. 2a, b). Importantly, DA-GIP-stimulated FA uptake into scWAT was reduced by muGIPR-Ab in both *Gipr*$^{fl/fl}$ and *Gipr*$^{βCell−/−}$ mice compared to *Gipr*$^{fl/fl}$ mice treated with DA-GIP only (Supplemental Fig. 2c), indicating that muGIPR-Ab inhibits GIP-stimulated FA uptake into scWAT independent of insulin secretion. Again, no effect was seen in eWAT or skeletal muscle (Supplemental Fig. 2d, e), but *Gipr*$^{βCell−/−}$ mice had significantly reduced liver FA uptake compared to *Gipr*$^{fl/fl}$ mice (Supplemental Fig. 2f), indicating that hepatic FA uptake may depend on the DA-GIP's incretin effect.

To assess how chronic muGIPR-Ab treatment affects systemic FA uptake, both lean and DIO mice were treated for 7 weeks. While lean mice did not have a treatment effect on body weight or tissue weights (Supplemental Fig. 2g–j), DIO mice treated with muGIPR-Ab had reductions in body weight (Supplemental Fig. 2g), scWAT weight (Supplemental Fig. 2h), and liver weight (Supplemental Fig. 2j), but not eWAT weight (Supplemental Fig. 2i) compared to vehicle-treated DIO mice. Consistent with the acute single-dose study (Fig. 4c, d), chronic muGIPR-Ab treatment inhibited FA uptake into scWAT (Supplemental Fig. 2k) without an effect on eWAT or skeletal muscle (Supplemental Fig. 2l, m). Again, consistent with the acute single-dose study (Fig. 4f), muGIPR-Ab-treated DIO mice had increased liver FA uptake compared to vehicle DIO mice (Supplemental Fig. 2n), this effect was reduced in DIO mice compared to lean mice, and notably, in the context of smaller livers, suggestive that the increased FA uptake does not lead to increased liver weight.

**Adipocyte GIPR partially contributes to muGIPR-Ab's effects.** Based on the direct actions of GIP on adipocytes and in scWAT specifically, we generated *Gipr* adipocyte knockout mice (*Gipr*$^{Adipo−/−}$) by crossing mice with a floxed *Gipr* gene (*Gipr*$^{fl/fl}$; mice previously described[2]) with mice expressing Cre recombinase driven by the adiponectin promoter (described in Methods

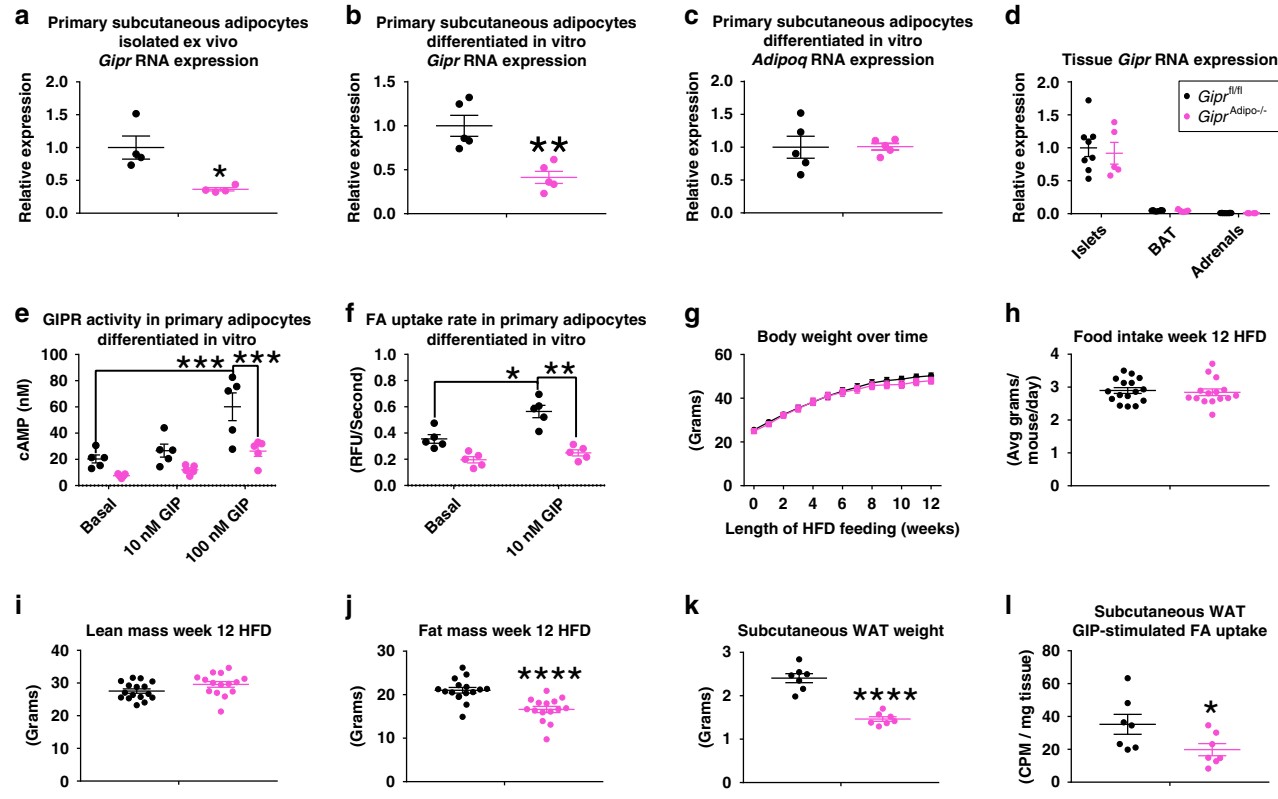

**Fig. 5 Mice with adipocyte *Gipr* knockout have reduced diet-induced fat mass accumulation. a** *Gipr* RNA expression in primary isolated adipocytes from scWAT from male *Gipr*^fl/fl and *Gipr*^Adipo−/− littermates fed HFD for 10 weeks; $n = 4$ mice/group, Student's paired $t$ test, *$p = 0.039$. **b** *Gipr* and **c** *Adipoq* RNA expression in primary mouse pre-adipocytes differentiated in vitro from male *Gipr*^fl/fl and *Gipr*^Adipo−/− littermates fed HFD for 18 weeks; $n = 5$ mice/group, Student's paired $t$ test, **$p = 0.0078$. **d** *Gipr* RNA expression in islets, brown adipose tissue (BAT), and adrenal glands relative to *Gipr*^fl/fl islets from male *Gipr*^fl/fl and *Gipr*^Adipo−/− littermates fed HFD for 18 weeks; $n = 8$ *Gipr*^fl/fl and $n = 5$ *Gipr*^Adipo−/−, Student's unpaired $t$-test. **e** GIP-stimulated GIPR activity in primary mouse pre-adipocytes differentiated in vitro from male *Gipr*^fl/fl and *Gipr*^Adipo−/− littermates fed HFD for 18 weeks; $n = 5$ mice, three wells/treatment/mouse, Two-way repeated measures ANOVA with Tukey's HSD for multiple comparisons, *Gipr*^fl/fl/Basal vs. *Gipr*^fl/fl/100 nM ***$p = 0.0003$ and *Gipr*^fl/fl/100 nM vs. *Gipr*^Adipo−/−/100 nM ***$p = 0.0009$. **f** FA uptake rate in primary differentiated mouse pre-adipocytes differentiated in vitro from male *Gipr*^fl/fl and *Gipr*^Adipo−/− littermates fed HFD for 18 weeks. $n = 5$ mice, three wells/treatment/mouse, two-way repeated measures ANOVA with Tukey's HSD for multiple comparisons, *$p = 0.022$, **$p = 0.0050$. **g–j** Male *Gipr*^fl/fl and *Gipr*^Adipo−/− littermates fed HFD for 12 weeks and **g** body weight measured over time, **h** food intake determined at week 12 HFD feeding, and **i** lean mass and **j** fat mass measured by MRI at week 12 HFD feeding. $n = 16$ *Gipr*^fl/fl mice and $n = 15$ *Gipr*^Adipo−/− mice, ****$p ≤ 0.0001$, **g** two-way repeated measures ANOVA with Sidak's test for multiple comparisons, **h–j** Student's unpaired $t$-test. **k, l** GIP-stimulated fatty acid uptake into scWAT of male *Gipr*^fl/fl and *Gipr*^Adipo−/− littermates fed HFD for 12 weeks. **k** ScWAT weight was measured at necropsy 180 mins post-dose and **l** radioactivity uptake into scWAT was determined. $n = 7$ mice/group, Student's unpaired $t$-test, ****$p ≤ 0.0001$ and *$p = 0.050$. **a–l** Data are presented as mean values ± SEM.

section and Supplemental Fig. 3a–f). After high-fat diet (HFD) feeding, *Gipr*^Adipo−/− mice had a significant reduction of *Gipr* RNA expression in primary adipocytes isolated ex vivo (Fig. 5a) and in primary pre-adipocytes differentiated in vitro (Fig. 5b) without an alteration in adipocyte *Adipoq* expression (Fig. 5c) or *Gipr* expression in other *Gipr* expressing tissues (Fig. 5d). GIPR activity was ablated in response to GIP-stimulated cAMP production (Fig. 5e) and GIP-stimulated FA uptake (Fig. 5f) in primary adipocytes differentiated in vitro. During HFD feeding, *Gipr*^Adipo−/− mice did not have differences in body weight (Fig. 5g), food intake (Fig. 5h), or lean mass (Fig. 5i) compared to their *Gipr*^fl/fl littermates. However, *Gipr*^Adipo−/− mice do have 20% less fat mass after 12 weeks HFD feeding (Fig. 5j) as reflected in significantly reduced scWAT (Fig. 5k). *Gipr*^Adipo−/− mice also had 45% lower GIP-stimulated scWAT FA uptake (Fig. 5l), indicating impaired adipocyte GIPR activity in vivo.

To understand the contribution of GIPR in adipocytes to the effect of muGIPR-Ab to prevent weight gain in DIO mice, *Gipr*^fl/fl and *Gipr*^Adipo−/− male littermates fed HFD for 12 weeks (described in Fig. 5g–j) were treated with vehicle or muGIPR-

Ab for 48 days. As expected, vehicle-treated *Gipr*^fl/fl mice gained 5% of their starting body weight over time and muGIPR-Ab treated mice had significantly reduced weight gain with a 2% reduction in their starting body weight over time, which significantly differed from vehicle from day 10 throughout the rest of the study (Fig. 6a). In *Gipr*^Adipo−/− mice, vehicle-treated mice gained 4.8% of their starting body weight, similar to *Gipr*^fl/fl mice, but muGIPR-Ab treated *Gipr*^Adipo−/− mice only had a reduction of 0.5% of their starting body weight, which never significantly differed from the vehicle-treated *Gipr*^Adipo−/− mice over time up to day 48 (Fig. 6b). Utilizing two-way repeated measures ANOVA, there is a significant treatment × genotype interaction where $p ≤ 0.0001$ for data represented in Fig. 6a, b.

To confirm the difference in the response of male *Gipr*^fl/fl and *Gipr*^Adipo−/− DIO littermates treated with muGIPR-Ab, we repeated treatment in a different model of DIO utilizing female *Gipr*^fl/fl and *Gipr*^Adipo−/− littermates fed HFD for 8 weeks then treated with muGIPR-Ab for 67 days. Vehicle-treated female *Gipr*^fl/fl mice gained 26.9% of their starting body weight while female *Gipr*^fl/fl mice treated with muGIPR-Ab only gained 8.1%

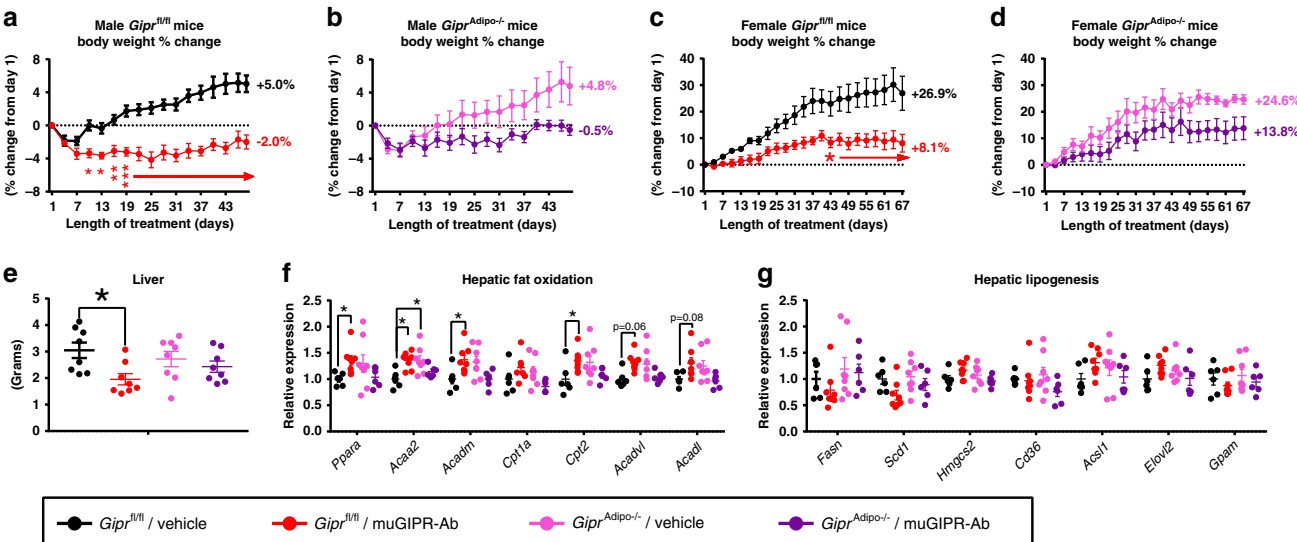

**Fig. 6 GIPR activity in adipocytes partially contributes to the effect of muGIPR-Ab to prevent weight gain in DIO mice.** Male **a** $Gipr^{fl/fl}$ and **b** $Gipr^{Adipo-/-}$ littermates fed HFD for 12 weeks (represented in Fig. 5g–j) were treated with vehicle or muGIPR-Ab (25 mg/kg every 6 days) for 48 days and body weight change measured over time. $n = 8$ mice/group $Gipr^{fl/fl}$/Vehicle, $Gipr^{fl/fl}$/muGIPR-Ab, and $Gipr^{Adipo-/-}$/Vehicle, and $n = 7$ mice $Gipr^{Adipo-/-}$/muGIPR-Ab; two-way repeated measures ANOVA with Sidak's test for multiple comparisons; **a** day 10 *$p = 0.028$, day 13 *$p = 0.035$, day 15 **$p = 0.0082$, and days 19–48 ****$p \leq 0.0001$. Data are presented as mean values ± SEM. Female (**c**) $Gipr^{fl/fl}$ and (**d**) $Gipr^{Adipo-/-}$ littermates fed HFD for 8 weeks were treated with vehicle or muGIPR-Ab (25 mg/kg every 6 days) for 67 days and body weight change measured over time. $n = 5$ mice/group $Gipr^{fl/fl}$/Vehicle, $Gipr^{fl/fl}$/muGIPR-Ab, and $Gipr^{Adipo-/-}$/muGIPR-Ab, and $n = 4$ mice $Gipr^{Adipo-/-}$/Vehicle; two-way repeated measures ANOVA with Sidak's test for multiple comparisons; **c** day 43 *$p = 0.027$, day 46 *$p = 0.018$, day 49-48 **$p \leq 0.0050$, days 61–67 ***$p \leq 0.008$. Data are presented as mean values ± SEM. **e** Liver weight from male $Gipr^{fl/fl}$ and $Gipr^{Adipo-/-}$ treated with vehicle or muGIPR-Ab for 48 days were collected at necropsy, one-way ANOVA with Sidak's test for multiple comparisons, *$p = 0.024$. Hepatic **f** fat oxidation and **g** lipogenesis genes were measured by Quantigene Plex. $n = 8$ mice/group $Gipr^{fl/fl}$/Vehicle, $Gipr^{fl/fl}$/muGIPR-Ab, and $Gipr^{Adipo-/-}$/Vehicle, and $n = 7$ mice $Gipr^{Adipo-/-}$/muGIPR-Ab; two-way ANOVA with Sidak's test for multiple comparisons; $Ppara$ *$p = 0.033$, $Acaa2$ *$p = 0.032$, $Acadm$ *$p = 0.025$, and $Cpt2$ *$p = 0.042$. Data are presented as mean values ± SEM.

of their starting body weight, a difference that showed significance from days 49 throughout 67 days (Fig. 6c). Vehicle-treated female $Gipr^{Adipo-/-}$ mice gained 24.6% of their starting body weight, similar to female $Gipr^{fl/fl}$ littermates, but female $Gipr^{Adipo-/-}$ mice treated with muGIPR-Ab gained 13.8% of their starting body weight, a difference that never significantly differed from vehicle-treated $Gipr^{Adipo-/-}$ mice throughout the 67 days (Fig. 6d). Utilizing two-way repeated measures ANOVA, there is a significant treatment × genotype interaction where $p \leq 0.0001$ for data represented in Fig. 6c, d. Taken together in two different models of DIO, this study demonstrates that mice with adipocyte knockout of $Gipr$ did not fully respond to the muGIPR-Ab, indicating a partial role for adipocyte GIPR for the effect of muGIPR-Ab, but still leaves other tissues or cell types to also partially contribute to the effect of muGIPR-Ab to prevent weight gain.

Since there was increased liver FA uptake observed in wild-type DIO mice pre-treated with muGIPR-Ab (Fig. 4f and Supplemental Fig. 2n), we utilized liver collected from the male $Gipr^{fl/fl}$ and $Gipr^{Adipo-/-}$ DIO littermates treated with vehicle or muGIPR-Ab for 48 days (Fig. 6a, b) where $Gipr^{fl/fl}$ mice treated with muGIPR-Ab had significantly reduced liver weight but $Gipr^{Adipo-/-}$ mice did not have differences in liver weight (Fig. 6e). Since chronic treatment of muGIPR-Ab leads to reduced liver weight (Supplemental Fig. 2j and Fig. 6e), but increased liver FA uptake (Supplemental Fig. 3n), we explored genes related to both hepatic FA oxidation and hepatic lipogenesis. Interestingly, livers from $Gipr^{fl/fl}$ mice treated with muGIPR-Ab had a modest, but significant increase in many genes associated with hepatic FA oxidation, including $Ppara$, $Acaa2$, $Acadm$, and $Cpt2$ (Fig. 6f), without alteration in genes associated with hepatic lipogenesis (Fig. 6g), suggesting that in mice

chronically treated with muGIPR-Ab, the reduced liver weight is associated with increased FA uptake and increased hepatic FA oxidation and is dependent on adipocyte GIPR activity.

**Chronic GIPR agonism desensitizes adipocyte GIPR activity.** Based on data previously demonstrating ligand-mediated GIPR desensitization in 3T3-L1 adipocytes differentiated in vitro[10,25], we hypothesized that chronic GIPR agonism desensitizes GIPR activity in adipocytes. Primary mouse adipocytes differentiated in vitro were treated with or without DA-GIP (1 µM) for 24 h, then stimulated with either mouse GIP or isoproterenol. Adipocytes incubated without DA-GIP demonstrated a significant GIP dose-dependent increase in cAMP concentration while adipocytes pre-incubated with DA-GIP had significantly reduced cAMP concentrations after subsequent GIP stimulation (Fig. 7a). Both pre-treatment with and without DA-GIP displayed similar maximal cAMP accumulation after isoproterenol treatment (Fig. 7a), indicating intact response to other cAMP stimuli. Next, we determined how long GIP-mediated GIPR desensitization persisted. Primary mouse adipocytes differentiated in vitro were treated with or without DA-GIP (1 µM) for 24 h, then stimulated with mouse GIP after different lengths of washout time. There was a time-dependent increase in cAMP production in adipocytes pre-treated with DA-GIP; however, these adipocytes did not fully regain their sensitivity to subsequent GIP stimulation even after a 48-h washout period (Fig. 7b). To further understand the time course of desensitization, primary mouse adipocytes were incubated with or without DA-GIP (1 µM) in a time course, then stimulated with mouse GIP, and even after 1 h of DA-GIP incubation, maximal responsiveness to GIP was blunted, but longer incubation further reduced the responsiveness with maximal inhibition observed at 24 h (Fig. 7c). To determine the dose

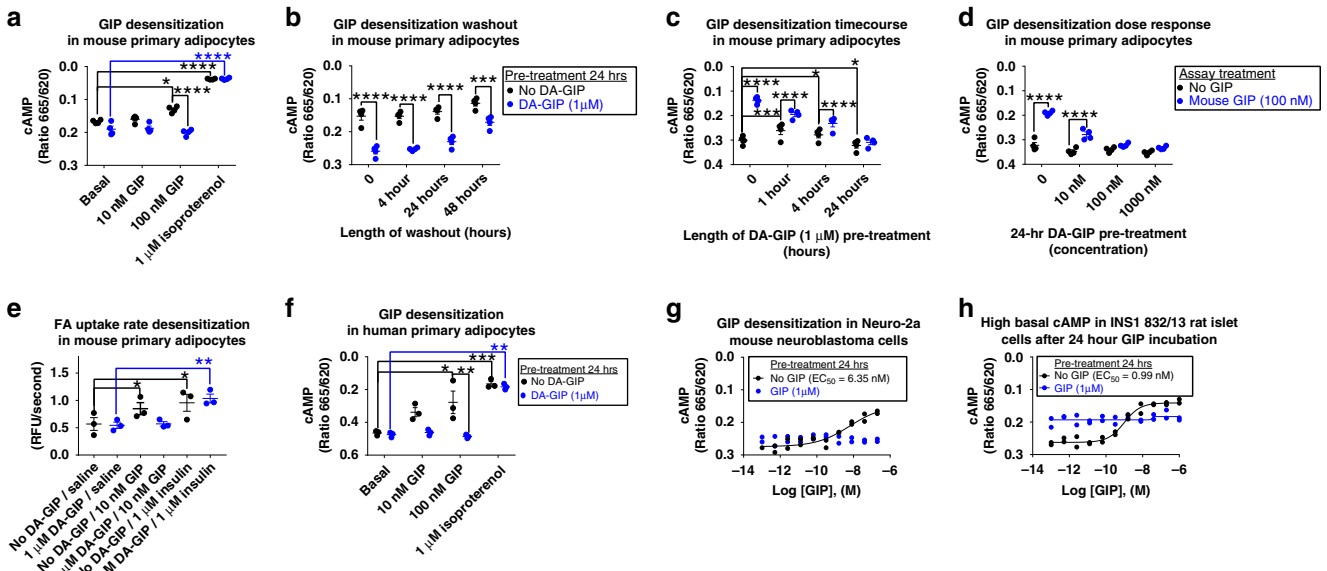

**Fig. 7 Chronic GIPR agonism desensitizes GIPR activity in primary adipocytes. a** cAMP measurements in primary differentiated mouse pre-adipocytes pre-treated with or without DA-GIP (1 μM) for 24 h then treated with vehicle, mouse GIP, or isoproterenol (1 μM) for 30 min; *$p = 0.010$, ****$p \le 0.0001$. **b** cAMP measurements in primary differentiated mouse pre-adipocytes pre-treated with or without DA-GIP (1 μM) for 24 h, washed, then treated with 100 nM mouse GIP for 30 min; ***$p = 0.0003$, ****$p \le 0.0001$. **c** cAMP measurements in primary differentiated mouse pre-adipocytes pre-treated with or without DA-GIP (1 μM) for 1, 4, or 24 h, then treated with 100 nM mouse GIP for 30 min; ****$p \le 0.0001$, ***$p = 0.0003$, 0/No GIP vs. 4 h/No GIP *$p = 0.021$, and 0/No GIP vs. 24 h/No GIP *$p = 0.027$. **d** cAMP measurements in primary differentiated mouse pre-adipocytes pre-treated with or without DA-GIP for 24 h, then treated with 100 nM mouse GIP for 30 min. ****$p \le 0.0001$. **e** FA uptake in primary differentiated mouse pre-adipocytes pre-treated with or without DA-GIP for 24 h, then treated with vehicle, mouse GIP, or insulin for 30 min. *$p = 0.015$, No DA-GIP Basal vs. No DA-GIP/Insulin **$p = 0.0044$, DA-GIP/Basal vs. DA-GIP/Insulin **$p = 0.0022$. **f** cAMP measurements in primary human adipocytes differentiated in vitro pre-treated with or without DA-GIP for 24 h then treated with vehicle, human GIP, or isoproterenol for 30 min. *$p = 0.011$, ***$p = 0.0009$, 100 nM GIP/No DA-GIP vs. 100 nM GIP DA-GIP **$p = 0.0065$, and Basal/DA-GIP vs. Isoproterenol/DA-GIP **$p = 0.0011$. **a**–**f** Data represent means ± SEM of four mice with three wells/treatment/mouse (**a**–**d**), three mice with three wells/treatment/mouse (**e**), three human donors with three wells/treatment/person (**f**); two-way repeated measures ANOVA with Sidak's multiple comparisons test. **g** Mouse Neuro-2a neuroblastoma cells and **h** rat INS1 832/13 islet cells pre-treated with mouse or rat GIP (1 μM), respectively, for 24 h then treated with mouse or rat GIP in a dose response for 15 min; $n = 2$ wells/treatment.

response of desensitization, primary mouse adipocytes were incubated with different concentrations of DA-GIP for 24 h, then stimulated with mouse GIP, and even 10 nM DA-GIP incubation blunted subsequent 100 nM mouse GIP stimulation with maximal inhibition observed at 100 nM and 1 μM DA-GIP pre-treatment (Fig. 7d).

To determine the functional consequences of GIPR desensitization in primary mouse adipocytes, they were pre-treated with or without DA-GIP (1 μM) for 24 h, then stimulated with mouse GIP or insulin and FA uptake rates were measured (Fig. 7e). Adipocytes pre-treated with or without DA-GIP then stimulated with insulin both had similar rates of FA uptake, whereas adipocytes pre-treated with DA-GIP did not respond to subsequent GIP stimulation seen in adipocytes without DA-GIP pre-treatment (Fig. 7e). When compared to the same study design utilizing muGIPR-Ab (Fig. 4b), chronic GIPR agonism functionally mimics GIPR antagonism in primary mouse adipocytes differentiated in vitro.

To confirm that GIP-induced receptor desensitization is not a mouse-specific phenomenon, we replicated the same experimental design in Fig. 7a in primary human adipocytes differentiated in vitro. Human adipocytes incubated without DA-GIP demonstrated a significant GIP dose-dependent increase in cAMP concentration while human adipocytes pre-incubated with DA-GIP (1 μM) had significantly reduced cAMP concentrations both with and without GIP stimulation and both conditions displayed similar maximal cAMP accumulation after isoproterenol treatment (Fig. 7f).

To understand GIPR desensitization in other cell types, we utilized mouse Neuro-2a neuroblastoma and rat INS1 832/13 islet cell lines. Mouse Neuro-2a cells had similar cAMP sensitivity as primary mouse adipocytes (Fig. 3m) to GIP ($EC_{50} = 6.35$ nM), and also similar to primary adipocytes, pre-treatment with GIP (1 μM) completely inhibited subsequent GIP stimulation (Fig. 7g). However, rat INS1 832/13 cell lines had a greater sensitivity to GIP ($EC_{50} = 0.99$ nM) than both primary mouse adipocytes and Neuro-2a neuroblastoma cells, and notably, pre-treatment with GIP (1 μM) for 24 h resulted in dramatically elevated basal cAMP levels that did not further respond to the subsequent GIP stimulation (Fig. 7h).

To investigate the molecular mechanisms by which ligand-induced GIPR desensitization occurs, we utilized overexpressing cell lines to detect membrane binding and employed imaging techniques, which cannot be done in endogenous cell lines that express relatively low levels of GIPR with current technology and reagents. First, cells overexpressing GIPR were pre-treated with or without DA-GIP (1 μM) then cells were washed and incubated with $^{125}$I-GIP for 2 h. Membrane bound $^{125}$I-GIP was ~50% lower in cells pre-treated with DA-GIP (Fig. 8a). Similarly, cells overexpressing GIPR were pre-incubated with or without DA-GIP (100 nM) for 24 h then incubated with Rhodamine-labeled GIP for 60 mins on ice, and membrane bound Rhodamine-GIP was also dramatically lower in cells pre-treated with DA-GIP (Supplemental Fig. 4a). Additionally, cells labeled with a SNAP Surface Alexa Fluor 647 substrate that only detects GIPR on the plasma membrane were treated with or without DA-GIP (100

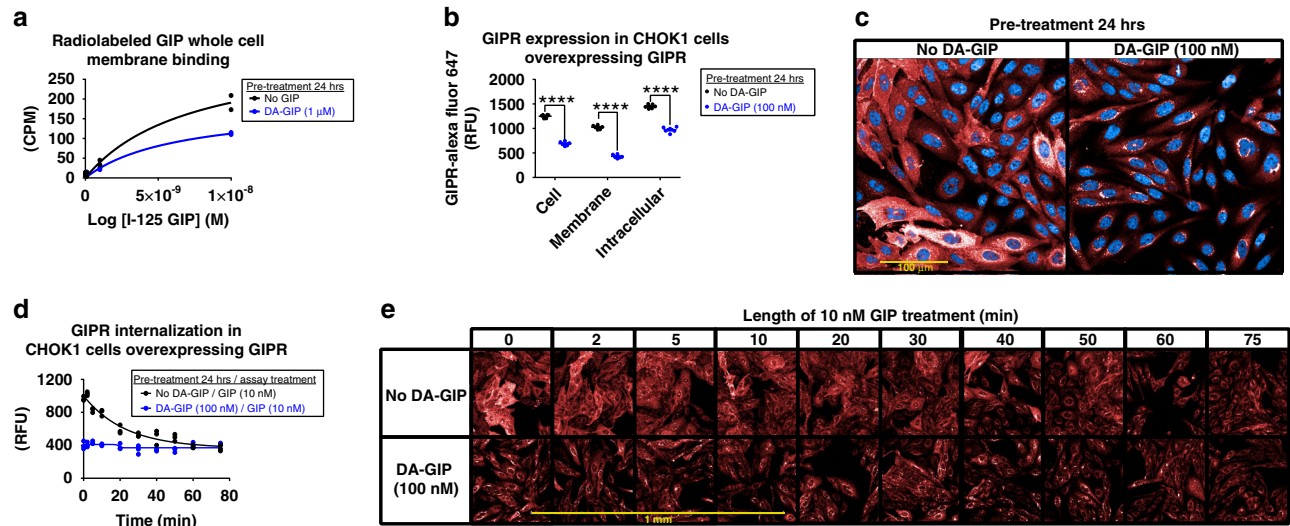

**Fig. 8 Chronic GIPR agonism increases membrane GIPR internalization in vitro. a** HEK293T cells overexpressing GIPR used to determine homologous whole cells $^{125}$I-GIP membrane binding. Cells treated with or without DA-GIP, washed, and binding using $^{125}$I-labeled human GIP was measure as CPM in a TopCount NXT gamma counter and plotted as % $^{125}$I-GIP bound. $n = 2$ wells/treatment. **b, c** Comparison of whole cell, membrane, and intracellular GIPR (red, Alexa Fluor 647) content in CHOK1 cells overexpressing GIPR pre-treated with or without DA-GIP for 24 h. **b** Quantitated GIPR fluorescence (RFU) and **c** representative image. $n = 6$ wells/treatment, two-way ANOVA with Sidak's test for multiple comparisons, ****$p \leq 0.0001$. Data are presented as mean values ± SEM. **d, e** Time course comparison of GIPR internalization upon stimulation with GIP in CHOK1 cells overexpressing GIPR (red, Alexa Fluor 647) pre-treated with or without DA-GIP for 24 h. **d,** Quantitated membrane GIPR fluorescence (RFU) with background fluorescence subtracted and **e**, representative images, $n = 3$ wells/treatment. Data are presented as mean values ± SEM.

nM) for 24 h, and cells incubated with DA-GIP had dramatically reduced plasma membrane SNAP-GIPR detection (Supplemental Fig. 4b, c). After subsequent re-stimulation with GIP (10 nM), cells that were not incubated with DA-GIP had significant internalization of SNAP-GIPR with disappearance of SNAP-GIPR from the plasma membrane and increased intracellular SNAP-GIPR content (Supplemental Fig. 4b, c). These plasma membrane binding and imaging studies suggest reduced availability of membrane GIPR after DA-GIP pre-treatment.

To confirm the hypothesis that DA-GIP pre-treatment reduces plasma membrane GIPR, cells overexpressing GIPR were pre-treated with or without DA-GIP (100 nM) for 24 h then cells were fixed throughout subsequent re-stimulation with GIP (10 nM) for imaging using a high content imaging analysis platform. After 24 h pre-treatment with DA-GIP, there was a significant reduction in total, plasma membrane, and cytoplasmic GIPR as quantified in Fig. 8b and imaged in Fig. 8c. After re-stimulation with GIP, cells without DA-GIP pre-treatment had reduced expression of membrane GIPR over time, indicating ligand-induced receptor internalization, which brought membrane GIPR levels down to a similar level as cells pre-treated with DA-GIP, and minimal to no additional GIPR internalization was observed in DA-GIP pre-treated cells (Fig. 8d, e). As previously reported in the literature[10,25], this data demonstrates ligand-mediated GIPR desensitization due to ligand-induced internalization that, at least in mouse primary adipocytes and Neuro-2a neuroblastoma cells, occurs within 24 h (Fig. 7d, g) and does not regain sensitivity for over 48 h (Fig. 7b), which explains why chronic GIPR agonism functionally mimics GIPR antagonism.

To assess the desensitization effects of the LA-Agonist treatment in vivo, DIO mice were treated with vehicle, LA-Agonist, or muGIPR-Ab for 6 days. Cumulative food intake was measured over time and both treatments equally reduced food intake compared to vehicle (Fig. 9a) as previously reported[2,3]. The LA-Agonist significantly reduced fasting blood glucose after 6 days while muGIPR-Ab did not (Fig. 9b) and neither treatment altered fasting insulin levels (Fig. 9c). There were no differences in

body weight, scWAT, eWAT, or liver weight between groups (Fig. 9d–g). As expected, DA-GIP significantly stimulated FA uptake into scWAT compared to saline and muGIPR-Ab blocked this effect; however, 6 days of treatment with the LA-Agonist also completely inhibited GIP-stimulated FA uptake into scWAT like the muGIPR-Ab (Fig. 9h) and no difference was observed in eWAT (Fig. 9i). Hepatic FA uptake was significantly reduced by GIP stimulation compared to saline (Fig. 9j), and no significant differences in skeletal muscle FA uptake was observed (Fig. 9k).

## Discussion

Here we demonstrate the in vivo and primary cell action of chronic GIPR agonism to desensitize GIPR and appear functionally like a GIPR antagonist. Chronic GIPR agonism and antagonism both provide similar efficacy regarding prevention of weight gain alone or weight loss when combined with GLP-1RAs as presented here and previously reported[2–5]. We hypothesized that this is explained by agonist-induced desensitization of GIPR where GIPR has a decreased response to an agonist upon repeated stimulation compared to the initial exposure to the same agonist[26]. Indeed, Mohammed et al.[10] demonstrated that an initial GIP stimulation can impair subsequent GIP stimulations associated with disappearance of GIPR from the plasma membrane in differentiated 3T3-L1 adipocytes. Importantly, it was shown that GIPR is constitutively internalized, i.e., receptor internalization even in the absence of GIP stimulation, which is an atypical characteristic of most GPCRs. Additionally, important work from another group with the peptidic antagonist GIP(3–30)NH$_2$ demonstrated that an antagonist actually restores expression of GIPR thus increases receptor surface expression[27]; however, we speculate that the increased surface expression of GIPR with GIP (3–30)NH$_2$ is functionally inactive owing to the presence of the peptide antagonist and thus renders the GIPR non-functional despite the increase in surface receptors.

We extended the work by the McGraw lab[10,25] and demonstrated that in primary mouse adipocytes differentiated in vitro, in primary human adipocytes differentiated in vitro, and in

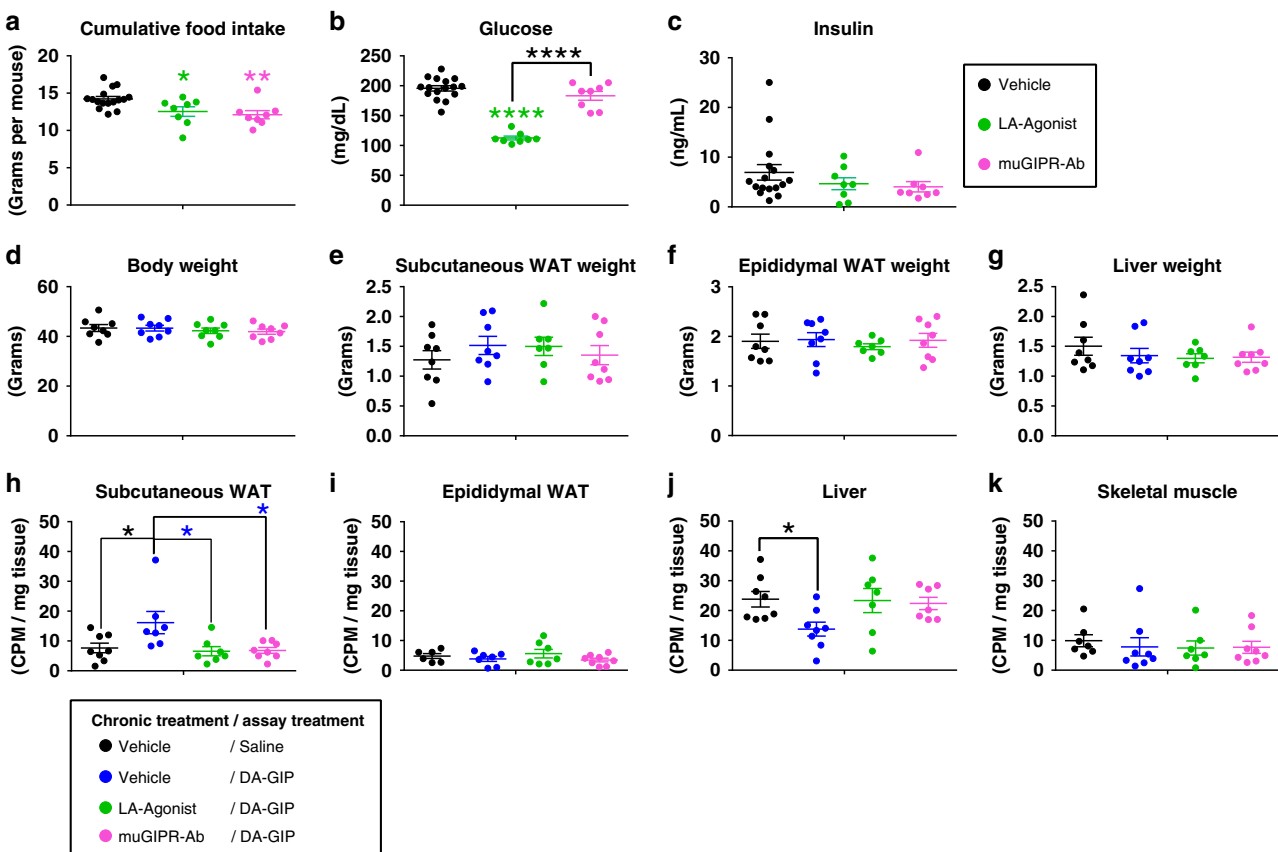

**Fig. 9 Chronic GIPR agonism desensitizes adipocyte GIPR activity in vivo mimicking GIPR antagonism.** GIP-stimulated FA uptake in DIO mice treated with vehicle, LA-Agonist, or muGIPR-Ab for 6 days. On day 6, mice were injected with saline or DA-GIP and simultaneously oral gavaged with $^{14}$C-oleic acid in olive oil to assess FA uptake. **a** Cumulative food intake determined; *$p = 0.036$, **$p = 0.0075$. **b** Fasting blood glucose and **c** plasma insulin were measured on day 6; ****$p \leq 0.0001$. Data represent means ± SEM of 16 mice for vehicle group (divided into two groups of 8 for the FA uptake assay) and eight mice/group for the LA-Agonist and muGIPR-Ab groups. **d** Body weight, **e** scWAT weight, **f** eWAT weight, and **g** liver weight were measured at necropsy on day 6. Radioactivity uptake into metabolically relevant tissues was determined at necropsy in **h** scWAT, **i** eWAT, **j** liver, and **k** skeletal muscle. **h** Vehicle/DA-GIP vs. Vehicle/Saline *$p = 0.027$, Vehicle/DA-GIP vs. muGIPR-Ab/DA-GIP *$p = 0.014$, Vehicle/DA-GIP vs. LA-Agonist/DA-GIP *$p = 0.014$; **j** *$p = 0.041$. $n = 8$ mice/group. **a–k** One-way ANOVA with Sidak's test for multiple comparisons; *$p \leq 0.05$, **$p \leq 0.01$, and ****$p \leq 0.0001$. Data are presented as mean values ± SEM.

mouse Neuro-2a neuroblastoma cells that chronic GIPR agonism impairs the ability of cells to produce cAMP in response to subsequent stimulation and this functionally results in the impairment of GIPR function to increase FA uptake into adipocytes. Using overexpressing cell lines, we confirmed by binding and imaging studies that membrane GIPR expression is dramatically reduced after chronic GIPR agonism due to receptor internalization. In vivo, both chronic GIPR agonism and antagonism resulted in decreased plasma cAMP, corticosterone, and glycerol, and resulted in impaired GIP-stimulated subcutaneous WAT FA uptake and reduction of food intake, a functional marker of centrally mediated GIPR activity, indicating GIPR desensitization in adipocytes, adrenal glands, and neuronal cells.

From the data demonstrating similarities in GIPR ligand-mediated desensitization in primary mouse adipocytes, primary human adipocytes, and mouse Neuro-2a neuroblastoma cell lines, we hypothesize that pharmacological GIPR agonism does not only desensitize GIPR in adipocytes, but may perhaps be a feature of cell types with relatively low levels of GIPR expression. Both the LA-Agonist and muGIPR-Ab display significantly reduced cumulative food intake after 6 days of treatment (Fig. 9a), indicating that ligand-mediated desensitization in neuronal cell types could potentially also occur as reflected by desensitization in mouse Neuro-2a neuroblastoma cells (Fig. 7g). This is also

supported by recent work from two different groups demonstrating that central administration of either GIP[28] and of an anti-GIPR antibody[29] both lead to a reduction in food intake, which we hypothesize can be effectively explained by desensitization of GIPR in neuronal cells by GIP leading to the same phenotype as an anti-GIPR antibody. Conversely, in a pancreatic islet cell line, rat INS1 832/13, cells appear to not desensitize, but rather display high basal levels of cAMP after chronic GIPR agonism (Fig. 7h), suggesting that GIPR continues to signal in these cells, although high basal activity with GIP pre-treatment and lack of response to fresh GIP represents desensitization to come extent, but the high basal levels, in contrast to the low levels in the other cell types, could reflect sustained activation, albeit not maximal. Similarly, mice treated with LA-Agonist for 21 days were able to respond to DA-GIP-stimulated insulin secretion 4 h after the last LA-Agonist dose (Fig. 2j), which suggests that GIPR in pancreatic β-cells does not desensitize in a manner observed adipocytes or neuronal cells. We hypothesize that cell type differences in desensitization can be explained by differences in GIPR expression where pancreatic islets have dramatically higher levels of *Gipr* expression compared to adipose tissue and brain sections (Supplemental Fig. 5). However, this is a hypothesis that requires further work to understand whether these cell type differences are due to differences in expression level or other components of cell signaling,

including β-arrestin recruitment and subcellular localization of the ligand-receptor complex. We now conclude that anti-GIPR antibodies do not require GIPR activity in pancreatic β-cells for efficacy[2], but rather partially depend on GIPR activity in adipocytes and further work is needed to find additional tissues or cell types for which GIPR therapies are dependent on, such as neuronal subpopulations as investigated by recent literature[28,29].

We and others have shown that GIP stimulates in vitro lipolysis in the absence of insulin[19]; however, physiological GIP concentrations likely do not act to promote lipolysis in vivo since the presence of high GIP will also stimulate insulin secretion, and insulin is a potent inhibitor of GIP-stimulated lipolysis[21]. Similar to insulin, GIP also stimulates FA uptake and re-esterification in rat adipose tissue[22,23] and in mouse primary adipocytes ex vivo[19,24]. Nevertheless, the mechanism of GIP-stimulated adipocyte lipolysis, rather than triglyceride storage, could potentially be an anti-obesity mechanism for pharmacological doses of GIPR agonists to overcome the anti-lipolytic effects of insulin. On the contrary, here we have demonstrated that in fact chronic GIPR agonism indeed decreases plasma glycerol, a marker of adipocyte lipolysis, as well as inhibits adipose tissue FA uptake, like a GIPR antagonist. Both acutely and chronically, the inhibition of adipose tissue FA uptake leads to an increase in liver FA uptake, and in chronic studies, results in reduced liver weight and increased liver FA oxidation genes, suggesting a redistribution of FA from storage in adipose tissue to oxidation in other tissues.

Additionally, it has been previously hypothesized that GIP's incretin action is responsible for decreased adiposity in *Gipr* knockout animals[30]. However, we have now shown that indeed GIP-stimulated insulin secretion is not required for FA uptake into adipose tissue in vivo and insulin co-incubation with GIP is not required for immediate GIP-stimulated FA uptake, which is not to say that insulin may not modulate or augment either of these effects acutely or chronically. Furthermore, mice with GIPR ablation in adipocytes had reduced adiposity after chronic HFD feeding, demonstrating regulation of adiposity, at least in part, by adipocyte GIPR. Overall, this work is important to understand the mechanism for clinical candidates targeting GIPR for the treatment of obesity, either agonists or antagonists, that chronically both inhibit the activity of GIPR at least in adipose tissue and potentially in others, such as adrenal glands or the brain.

**Limitations of study.** An important caveat is that these in vivo studies were performed in mice while other GIPR antagonists have been assessed in humans. GIP(3–30)NH$_2$ is a well described potent antagonist against human GIPR, and has been used effectively in human studies to prevent GIP-induced insulin secretion[31] and triglyceride uptake in adipose tissue by altering adipose tissue blood flow[32]. GIP(3–30)NH$_2$ binds rodent GIPR differentially (10–40 fold lower binding affinity as determined by a radioligand competition assay[27]), possibly owing to the sequence divergence of the rodent and human GIP peptides. Despite the effectiveness of the GIP(3–30)NH$_2$ in human, its poor half-life limits its utility as both a pharmacological agent and a tool compound for preclinical weight loss studies. We interpret these species differences to possibly relate to sequence divergence of the GIP ligand and the GIPR extracellular domain between rodents and humans and may not represent an intrinsic difference in GPCR activation and G-protein recruitment. Based on our previous work using a human anti-GIPR antibody in efficacy studies in nonhuman primates that had improved efficacy on weight loss compared to muGIPR-Ab in DIO mice[2], we think the use of a mouse-specific antagonistic antibody to GIPR (muGIPR-Ab) will translate to human studies, particularly since the GIP

EC$_{50}$ for cAMP in mouse primary adipocytes (3.8 nM, Fig. 3m) and in human primary adipocytes (1.3 nM[2]) are similar, and here we have demonstrated identical ligand-induced GIPR desensitization in both mouse and human primary adipocytes (Fig. 7a, f). However, differences in responsiveness between obese and diabetic individuals exist that cannot be accounted for by utilizing DIO mice, though as shown in Supplemental Fig. 2g–n, lean mice do not differ in body weight, adipose tissue or liver weight, or GIP-stimulated FA uptake after treatment with muGIPR-Ab as seen in DIO mice, suggesting inherent differences in GIPR signaling between lean and obese animals.

Additionally, here we focused on the direct effects of the LA-Agonist and muGIPR-Ab alone and not in combination with GLP1-RAs. We feel that this is important to understand the biology of GIPR alone but recognize that co-administration of GLP-1RAs could modulate the effects presented here either directly or indirectly, though adipocytes do not express GLP-1R. Moreover, many of the in vivo assays here used super-physiological concentrations of DA-GIP to develop these assays and may not reflect physiological activity but do remain important for considering the effects of GIPR agonists in vivo.

## Methods

**LA-agonist generation.** Starting with the (G4S)$_3$K(ivdde)-Rink amide resin sub-assembly (0.6 mmol, CEM), the remainder of the peptide (sequence: Y[Aib]EGT-FISDYSIAMDKIHQQDFVNWLLAQKGKKNDWKHNITQ) was built using DIC/Oxyma chemistry in DMF on a Protein Technologies Tribute peptide synthesizer and standard reagents for Fmoc-SPPS. For the final coupling in the sequence, Boc-Y(OtBu)-OH was utilized. For each coupling, 5 eq of DIC, Oxyma, and amino acid were used. Most couplings were performed at 75 °C with IR heating for 5 min (final Boc-Y coupling was performed at 50 °C for 25 min, both H couplings were performed at 50 °C for 10 min). Double-couplings were performed after each aromatic or beta-branched residue. Deprotections were performed at 50 °C with 20% 4-Me-piperidine in DMF.

After completion of the main chain, the resin was placed in a 100-mL synthesis vessel and washed with DMF (30 mL x2). The resin was treated with 5% Hydrazine in DMF (10 mL) and gently shaken at room temperature for 5 min. The solution was drained under vacuum and the deprotection was repeated 10 times. The resin was washed with DMF (100 mL x2) and DCM (100 mL x3). In a separate vessel, a solution of DIC (0.935 mL, 6.00 mmol) in DMF (10 mL) was slowly added to a solution of bromoacetic acid (1.667 g, 12.00 mmol) in DCM (30 mL) (exotherm produced). The mixture was stirred at room temperature for 10 min then added to the previously deprotected resin. The reaction was shaken at room temperature for 18 h, then the solvent was drained, and the resin was washed with DCM (50 mL x4).

The resin was treated with TIPS (2 mL), water (2 mL), and TFA (40 mL). The mixture was kept at room temperature for 3 h with stirring. The cleavage solution was drained into a 250-mL RBF and concentrated to about half of the original volume on rotavap. Cold cyclopentylmethyl ether was added to the solution and the suspension was filtered and washed with more CPME yielding 2.55 g of crude material.

The crude peptide was dissolved in Water/MeCN (2:1) and injected onto a prep HPLC column (Phenomenex Gemini, 5 μ, 250 ×30 mm (~300 mg/injection), and A: 0.1% TFA in H$_2$O, B: 0.1% TFA in CH$_3$CN). The column was eluted with a gradient of 20–40%B over 50 min with a 5-min flush and 10-min equilibration at a flow rate of 60 mL/min. The fractions were analyzed by LC-MS. The purification was repeated 10 times and the combined pool was lyophilized to give dry peptide. The peptide was dissolved in water and submitted for cation exchange purification (GE HiPrep 16/10 SP/HP column, 5 mL/min, A: 20 mM NaOAc, pH 5.0, B: 20 mM NaOAc,1.0 M NaCl, pH 5.0, Gradient A– >B 0–50% over 50 min). The fractions were desalted using Amicon Ultra-15 spin concentrators (MWCO = 3000) into water before being freeze-dried by lyophilization, yielding 136 mg of pure Y[Aib]EGTFISDYSIAMDKIHQQDFVNWLLAQKGKKNDWKHNITQGGGGSGGGGSGGGGSK(BrAc)-NH$_2$. (71% pure by HPLC, m/z predicted: 1032.92; 1239.30; 1548.88, found: 1033.0, 1239.2, 1548.5)

Serum-free adapted CHOK1 cells were transfected with plasmid DNA using a lipid-based transfection reagent. After antibiotic selection, a stable pool expressing the monoclonal antibody was propagated in proprietary in-house media. A 10-day fed-batch production was carried out in GE Wave bioreactor. In a separate flask, 100 ml GenScript AmMag Protein A Magnetic Beads are sanitized with 0.2 N sodium hydroxide, equilibrated and then slurried at 1:1 ratio in PBS, and finally added to the 25 L Wave Bag ~18 h prior to harvesting. The following day, the magnetic Protein A beads are trapped in the Wave Bag with a plate magnet while the cells and spent media are drained. The 100 ml of Protein A beads are then

resuspended in 2.5 L of PBS and drained into a bottle as the starting material for purification.

The Genscript AmMag Protein A Magnetic Beads in PBS are poured into a 10-L Conical Tank customized for magnetic bead processing. Magnetic beads are held within the tank with a ring magnet and the PBS is drained. Magnetic beads are then resuspended and washed 2x with 2.5 L of de-ionized water. The beads are next washed 3x with 2 L of 25 mM TRIS-HCl, 0.1 M sodium chloride, pH 7.4 (TBS), and followed by a final 5-L de-ionized water wash to remove all buffering components of the TBS in preparation for the elution step.

To elute the antibody, the beads are rinsed from the tank with 200 ml of 0.1 M sodium acetate, pH 3.6 into a 500-ml bottle and placed on a roller rack at 65 rpm for 15 min. The beads are held in the bottle with a small circular magnet while the elution is poured into another bottle. The elution is immediately neutralized with a 1% (v/v) addition of 3 M Tris Base. The elution step is repeated 3x to ensure full antibody elution from Protein A beads. The material was buffer exchanged into 10 mM sodium acetate, 9% sucrose, and pH 5.2. Heavy chain sequence (conjugation site in bold font): QVQLVESGGGVVQPGRSLRLSCAASGFTFSSYGMH WVRQAPGKGLEWVAVIWYDGSNKYYADSVKGRFTISRDNSKNTLYLQMNS LRAEDTAVYYCARYNFNYGMDVWGQGTTVTVSSASTKGPSVFPLAPSSKST SGGTAALGCLVKDYFPEPVTVSWNSGALTSGVHTFPAVLQSSGLYSLSSVVT VPSSSLGTQTYICNVNHKPSNTKVDKKVEPKSCDKTHTCPPCPAPELLGGPSV FLFPPKPKDTLMISRTPEVTCVVVDVSHEDP**C**VKFNWYVDGVEVHNAKTKP CEEQYGSTYRCVSVLTVLHQDWLNGKEYKCKVSNKALPAPIEKTISKAKGQ PREPQVYTLPPSREEMTKNQVSLTCLVKGFYPSDIAVEWESNGQPENNYKT TPPVLDSDGSFFLYSKLTVDKSRWQQGNVFSCSVMHEALHNHYTQKSLSLS PGK.

Light chain sequence: DIQMTQSPSSVSASVGDRVTITCRASQGISRRLAWY QQKPGKAPKLLIYAASSLQSGVPSRFSGSGSGTDFTLTISSLQPEDFATYYCQQ ANSFPFTFGPGTKVDIKRTVAAPSVFIFPPSDEQLKSGTASVVCLLNNFYPREA KVQWKVDNALQSGNSQESVTEQDSKDSTYSLSSTLTLSKADYEKHKVYAC EVTHQGLSSPVTKSFNRGEC.

To a 500-mL bottle was added to 300 mL of 2.5 mM cystamine, 2.5 mM cysteamine, 40 mM HEPES, pH 8.2, and an antibody with engineered cysteines (Cys-mAb) (197.5 ml, 10.9 mg/mL, and 0.015 mmol). The mixture was shaken for 18 h at room temperature. The solution was filtered through a 0.2-μm filter and diluted with 100 mM NaOAc, pH 5.0, then loaded directly onto the CEX column. The mixture was purified by cation exchange chromatography (GE 240 mL SP/HP, A: 20 mM NaOAc, pH 5.0, B: A + 1.0 M NaCl, 0–30% over 10 CV, and 20 mL/ min). The fractions containing product were combined and buffer exchanged by TFF (Millipore Pellicon3, Ultracel 30 kDa Membrane, 88 cm$^2$) into 10 mM sodium acetate, 9% sucrose, pH 5.2, yielding 450 mL of a solution of Cys-mAb-CA cap at 4.3 mg/mL (90% yield). LC-TOF MS: Predicted 144510, found: 144485.

To a 1000-mL reservoir on a TFF instrument was added 10 mM sodium acetate, 9% sucrose, pH 5.2 (100 mL), Cys-mAb-CA cap (232 ml, 6.88 μmol) and triphenylphosphine-3,3′,3″-trisulfonic acid trisodium salt (Sigma-Aldrich, 3.4 mM in H$_2$O; 8.09 ml, 0.028 mmol). The mixture was gently stirred for 90 min at room temperature. When the reaction was complete by ion exchange chromatography, the solution was concentrated to 100 mL and buffer exchanged by TFF (30 kD MWCO, Millipore Ultracel membrane, 88 cm$^2$) into 10 mM sodium acetate, 9% sucrose, pH 5.2. To the solution was added 150 mL of 50 mM sodium phosphate, 2 mM EDTA, pH 7.5, followed by dehydroascorbic acid (Biosynth International, 4.0 mM in 50 mM sodium phosphate, 2 mM EDTA, pH 7.5; 17.20 ml, 0.069 mmol). The re-oxidation was monitored by HPLC to minimize the amount of free light chain species in relation to fully intact Ab. When the reaction was sufficiently complete, GIPR agonist peptide (2.0 mM in H$_2$O; 20.64 ml, 0.041 mmol) was added. The mixture was gently stirred at room temperature for 18 h. The solution was concentrated to 100 mL and buffer exchanged into A5Su using TFF (MWCO 30,000). The concentrated solution was recovered from the instrument with several washes. The solution was filtered through a 0.22-μm filter and diluted with 2 M ammonium sulfate, 20 mM NaOAc, pH 5.0. The solution was purified using hydrophobic interaction chromatography (Custom GE Sepharose butyl HP column, 159 mL, ~35 mm × 200 mm). The solution was directly loaded onto the column with a sample loading pump. The desired compound was eluted under gradient conditions (12 mL/min, 100% A for 2 CV, 0–90% over 10 CV, hold for 2 CV, 100% for 2 CV, pure water for 2 CV, A: 1 M ammonium sulfate, 20 mM NaOAc, pH 5.0, B: 20 mM NaOAc, 10% CH$_3$CN, pH 5.0). The fractions containing DAR2 were pooled, combined with identical product runs, concentrated, and buffer exchanged into 10 mM sodium acetate, 9% sucrose, pH 5.2 using TFF (30 kD MWCO, Millipore Ultracel membrane, 88 cm$^2$). The purified product was collected with washes to give 302 mL of a 3.4-mg/mL solution. The final concentration was achieved by centrifugal ultrafiltration using Amicon Ultra-15 spin concentrators (30000 MWCO) followed by filtration through a 0.22-μm syringe filter to give 52 mL of 19.4 mg/mL LA-Agonist solution (50% yield overall). 96.1% purity by SEC, LC-TOF MS: predicted: 156,580, found: 156,574.

**LA-Agonist in vitro selectivity.** LA-Agonist activity was assessed in vitro in CHO cells stably expressing mouse GIPR, mouse GLP1R, or human glucagon receptor compared to mouse GIP (Phoenix), mouse GLP-1 (Phoenix), mouse glucagon (Phoenix), human DA-GIP (Phoenix), and LA-Agonist peptide only. Briefly, cAMP production was determined using a homogeneous time-resolved

fluorescence (HTRF) assay (Cisbio). Serial diluted agonists were incubated with 40,000 cells in assay buffer (0.1% BSA and 500 μM IBMX in DMEM) for 15 min at 37 °C. Cells were then lysed with lysis buffer containing cAMP-d2 and cAMP cryptate (Cisbio, Bedford, Massachusetts) and incubated for 1 h at room temperature before measurement in the Envision plate reader (PerkinElmer, Waltham, Massachusetts). The cAMP levels are expressed as a fluorescence ratio of 665/620 nm.

**LA-Agonist in vivo studies.** All studies using mice complied with all relevant ethical regulations and approval for the studies performed were obtained from Amgen's IACUC institutional review board (Thousand Oaks, CA). Lighting in animal holding rooms was maintained on 12:12 h light:dark cycle, and the ambient temperature and humidity range was at 68–79 °F and 30–70%, respectively. Animals had ad libitum access to irradiated pelleted feed and reverse-osmosis chlorinated (0.3–0.5 ppm) water via an automatic watering system. Cages were changed weekly. Male C57Bl6/J DIO mice (stock #380050) or age-matched lean controls (stock #380056) were purchased from Jackson laboratory. After arrival, DIO mice were continued on high-fat diet (HFD; 60 kcal% fat, Research Diets) or lean controls fed standard chow diet (Envigo Teklad Global Rodent Diet-soy protein-free extruded 2020X).

To assess in vivo selectivity, male DIO Gipr$^{fl/fl}$ and Gipr$^{βCell−/−}$ littermates (mice previously described[2]) were fasted for 6 h, then baseline retroorbital (RO) bleed taken (T0), and immediately IP injected with glucose (0.5 g/kg) and saline, DA-GIP (250 nmol/kg), or LA-Agonist peptide only (250 nmol/kg) as indicated then blood samples collected by RO bleed after 30 mins. Plasma insulin was measured using the High Range Mouse Insulin ELISA (Alpco).

LA-Agonist pharmacodynamic (PD) assay for GIP-stimulated insulin secretion where DIO mice fed HFD for 13 weeks were IP injected with saline or LA-Agonist in escalating doses, fasted for 4 h, then baseline retroorbital bleed taken (T0), and immediately IP injected with saline or DA-GIP (50 nmol/kg) and 2 g/kg glucose then blood samples collected by RO bleed over time for blood glucose measured by glucometer and plasma insulin. Plasma insulin was measured using the High Range Mouse Insulin ELISA (Alpco).

LA-Agonist was administered as a single intravenous (IV) or IP injection of 5 mg/kg to male CD-1 mice (n = 3 mice/administration route/timepoint). Blood samples for PK analysis were collected at serial time points up to 7 days post-dose. Samples were processed to plasma and stored at ~−70 °C (±10 °C) until transferred for subsequent analysis.

LA-Agonist concentrations in mouse plasma were determined by liquid chromatography tandem mass spectrometry (LC-MS/MS). LA-Agonist stock solution (1 mg/mL) was made from reference standards in 10 mM sodium acetate, 9% sucrose, pH 5 (A5Su) buffer and used to prepare a 100-μg/mL working solution in A5Su buffer. Standard concentrations of 250, 500, 1000, 2500, 5000, and 10,000 ng/mL were prepared by serial dilution of a freshly prepared 10,000 ng/mL solution in mouse plasma using the 100-μg/mL LA-Agonist working solution. In all, 25 μl mouse plasma samples were aliquoted into the appropriate well of a 96-well plate, followed by immunoaffinity capture using Protein-A cartridges on an AssayMap Bravo affinity purification system (Agilent). Samples were then eluted with 25 μl of 50% ACN 0.1% formic acid. The eluted samples were denatured using 0.1% RapiGest (Waters) and reduced by tris(2-carboxyethyl) phosphine (TCEP) and followed by trypsin digestion. After quenching with formic acid, samples were transferred to a 96-well plate. LC-MS/MS analysis was performed utilizing an ACQUITY UPLC system (Waters) and a Sciex 5500 QTrap MS system, the raw data were collected using Sciex Analyst® v1.5. LC-MS/MS detection of intact LA-Agonist was performed using a surrogate peptide of the GIP peptide N-terminus with a sequence of Y[Aib]EGTFISDYSIAMDK. The concentrations of plasma quality controls (QCs) and unknown samples were calculated using Watson (v7.4; Thermo) by weighted linear regression with a weighting factor of 1/x$^2$ from plasma calibration standards run within the same batch.

LA-Agonist concentrations in mouse plasma samples collected during the PD assay were determined as described above. The IC$_{50}$ and EC$_{50}$ of LA-Agonist in relation to the glucose and insulin response, respectively, was calculated following logarithmic transformation and nonlinear fit of data using a sigmoidal dose-response model with variable slope in GraphPad Prism (v7.02; GraphPad Software).

LA-Agonist PK was initially characterized utilizing noncompartmental analysis (NCA) of observed data from the single-dose PK study in mice. Plasma concentration-time data were subsequently fit by nonlinear regression to alternative compartmental PK models, with model selection guided by goodness-of-fit diagnostics and parameter estimate precision. A 2-compartment PK model with first-order elimination was found to best describe the LA-Agonist plasma data. Ensuing PK modeling was performed to simulate LA-Agonist exposure and predict dosing necessary to achieve target coverage in the chronic efficacy study. PK parameter estimates generated from NCA are provided in Supplemental Information (Supplemental Table 1). All PK analyses were conducted in Phoenix WinNonlin (v6.4; Certara).

DIO mice fed HFD for 12 weeks at the start of the study dosed with vehicle (saline 1x/day and vehicle 1x every 6 days), liraglutide (0.3 mg/kg liraglutide 1x/day and vehicle 1x/day; Bachem), LA-Agonist (37.5 mg/kg LA-Agonist 1x/day and saline 1x/day), muGIPR-Ab (25 mg/kg muGIPR-Ab 1x every 6 days and saline 1x/

day), LA-Agonist + liraglutide (37.5 mg/kg LA-Agonist 1x/day and 0.3 mg/kg liraglutide 1x/day), and muGIPR-Ab + liraglutide (25 mg/kg muGIPR-Ab 1x every 6 days and 0.3 mg/kg liraglutide 1x/day) for 21 days. MRI measurement taken at treatment days −2 and 18, and body weight and food intake measured over time. On day 21, regular dosing followed by 4-h fast, then T0 RO bleed immediately followed by IP injections of DA-GIP (50 nmol/kg) + glucose (2 g/kg) given to all mice with a RO bleed at T15 and terminal decapitation blood collection at T80 min, then tissues collected, weighed, and flash frozen. Plasma insulin was measured by ELISA (Alpco) and plasma adipokines were measured using Milliplex MAP Mouse Adipokine Magnetic Bead Panel (Millipore Sigma).

**Plasma metabolomics.** Plasma metabolomics was performed on terminal decapitation blood collected 80 min post-glucose and DA-GIP injection as described above by Metabolon using the Metabolon HD4 Platform. Plasma samples were prepared using the automated MicroLab STAR® system from Hamilton Company. Several recovery standards were added prior to the first step in the extraction process for QC purposes. Samples were extracted with methanol under vigorous shaking for 2 min (Glen Mills GenoGrinder 2000) to precipitate protein and dissociate small molecules bound to protein or trapped in the precipitated protein matrix, followed by centrifugation to recover chemically diverse metabolites. The resulting extract was divided into five fractions: two for analysis by two separate reverse phase (RP)/UPLC-MS/MS methods using positive ion mode electrospray ionization (ESI), one for analysis by RP/UPLC-MS/MS using negative ion mode ESI, one for analysis by HILIC/UPLC-MS/MS using negative ion mode ESI, and one reserved for backup. Samples were placed briefly on a TurboVap® (Zymark) to remove the organic solvent. The sample extracts were stored overnight under nitrogen before preparation for analysis. Several types of quality control samples were analyzed in concert with the experimental samples. These include: (1) technical replicate samples derived from a pool of well-characterized human plasma (MTRX), (2) extracted water samples (process blanks) and solvent blanks; and (3) a cocktail of QC standards, carefully chosen not to interfere with the measurement of endogenous compounds, spiked into every analyzed sample, allowing instrument performance monitoring and aiding with chromatographic alignment.

Ultrahigh Performance Liquid Chromatography-Tandem Mass Spectroscopy (UPLC-MS/MS): All methods utilize a Waters ACQUITY ultra-performance liquid chromatography (UPLC) and a Thermo Scientific Q-Exactive high resolution/accurate mass spectrometer interfaced with a heated electrospray ionization (HESI-II) source and Orbitrap mass analyzer operated at 35,000 mass resolution. The sample extract was dried then reconstituted in solvents compatible to each of the four methods. Each reconstitution solvent contains a series of standards at fixed concentrations to ensure injection and chromatographic consistency. One aliquot was analyzed using acidic positive ion conditions, chromatographically optimized for more hydrophilic compounds. In this method, the extract was gradient eluted from a C18 column (Waters UPLC BEH C18–2.1×100 mm, 1.7 μm) using water and methanol, containing 0.05% perfluoropentanoic acid (PFPA) and 0.1% formic acid (FA). A second aliquot was also analyzed using acidic positive ion conditions but is chromatographically optimized for more hydrophobic compounds. In this method, the extract is gradient eluted from the aforementioned C18 column using methanol, acetonitrile, water, 0.05% PFPA and 0.01% FA, and was operated at an overall higher organic content. A third aliquot was analyzed using basic negative ion optimized conditions using a separate dedicated C18 column. The basic extracts are gradient eluted from the column using methanol and water, however with 6.5 mM Ammonium Bicarbonate at pH 8. The fourth aliquot was analyzed via negative ionization following elution from a HILIC column (Waters UPLC BEH Amide 2.1 × 150 mm, 1.7 μm) using a gradient consisting of water and acetonitrile with 10 mM Ammonium Formate, pH 10.8. The MS analysis alternates between MS and data-dependent MSn scans using dynamic exclusion. The scan range varies slightly between methods, but covers ~70–1000 m/z.

*Bioinformatics:* the informatics system consists of four major components, the Laboratory Information Management System (LIMS), the data extraction and peak-identification software, data processing tools for QC and compound identification, and a collection of statistical, visualization, and interpretation tools for use by data analysts. The hardware and software foundations for these informatics components are the LAN backbone and database servers running Oracle 10.2.0.1 Enterprise Edition.

*Data extraction and compound identification:* raw data were extracted, peak-identified, and QC processed using Metabolon's hardware and software, Compounds were identified by comparison to library entries of purified standards or recurrent unknown entities. Metabolon maintains a library based on authenticated standards that contains the retention time/index (RI), mass to charge ratio (*m/z*), and chromatographic data (including MS/MS spectral data) on all molecules present in the library. Furthermore, biochemical identifications are based on three criteria: retention index within a narrow RI window of the proposed identification, accurate mass match to the library ±10 ppm, and the MS/MS forward and reverse scores. MS/MS scores are based on a comparison of the ions present in the experimental spectrum to ions present in the library entry spectrum.

*Curation:* a variety of curation procedures are performed to ensure that a high-quality data set is made available for statistical analysis and data interpretation. The QC and curation processes are designed to ensure accurate and consistent identification of true chemical entities, and to remove those representing system

artifacts, mis-assignments, redundancy, and background noise. Metabolon data analysts use internally developed visualization and interpretation software to confirm the consistency of peak identification among the various samples. Library matches for each compound are checked for each sample and corrected if necessary. Peaks are quantified as area-under-the-curve detector ion counts.

*Statistical analysis:* all statistical analysis was performed by Metabolon, who were blinded to the experimental treatments, and were not altered in any way by Amgen or the authors. Standard statistical analyses were performed in ArrayStudio on log transformed data using Welch's two-sample *t*-test to test whether two unknown means are different from two independent populations with $p \le 0.05$ indicating statistical significance. To correct for multiple comparisons, the *q*-value method for False Discovery Rate was used at a cutoff $q \le 0.2$ [33].

*Pathway enrichment analysis:* pathway enrichment was determined using Metabolync software (Metabolon) where Enrichment Score = $(k/m)/(n/N)$, where $k$ = number of significant metabolites in a pathway, $m$ = total number of detected metabolites in the pathway, $n$ = total number of significantly different metabolites, and $N$ = total number of detected metabolites per pathway.

**In vivo GIP-stimulated corticosterone and glycerol secretion.** DIO mice fed HFD for 12 weeks fasted for 10 h overnight then at 7 a.m., blood sample collected by RO bleed then immediately IP injected with saline or DA-GIP in escalating doses, and blood collected by RO bleed over time as indicated. Plasma metabolites corticosterone (Alpco) and glycerol (Sigma) measured over time. DA-GIP EC₉₀ for corticosterone secretion was calculated determined following nonlinear fit of data using a sigmoidal dose-response model with variable slope (GraphPad Prism). Subsequently, DIO mice fed HFD for 12 weeks were pre-treated with vehicle or muGIPR-Ab 24-h before baseline bleed collection (T0), fasted for 10 h overnight then at 7 a.m., blood sample collected by RO bleed then immediately injected with saline or DA-GIP (80 nmol/kg), and blood collected by RO bleed as indicated. Plasma metabolites corticosterone and glycerol measured at T0 and 30 min post-dose.

**Primary mouse pre-adipocyte isolation and differentiation.** Using the method originally described by Viswanadha and Londos[34], the subcutaneous WAT was isolated and dissected from male DIO mice, weighed, and immediately submerged in Krebs-Ringer bicarbonate (KRB) buffer at pH 7.4 with 4% bovine serum albumin (BSA), 500 nM adenosine, and 5 mM glucose, and the stromal vascular fraction (SVF) and primary adipocytes were separated by collagenase digestion (1 mg/mL KRB) and incubated at 37 °C with shaking at 220 rpm for 1 h. After digestion, the mixture was filtered through a 250-μm gauze mesh into a 15-ml conical polypropylene tube and the infranatant containing the collagenase solution and the SVF was carefully removed using a long needle and syringe. The SVF was cultured as previously described by Hausman et al.[35] where the SVF containing solution was centrifuged at 200×g for 10 min to pellet the SVF cells, resuspended in 10 mL plating medium (DMEM/F12 + 10% FBS), then filtered through a sterile 20-μm mesh filter into a sterile 50-mL plastic centrifuge tube. SVF cells were plated in 24-well plate at 250,000 cells/well and incubated at 37 °C and 5% CO2 overnight then the plating medium and nonadherent cells were removed, replaced with DMEM/F12 media + 5% FBS, and media was replaced every two days until cells reached confluency (5–6 days after plating). Differentiation was induced by the addition of differentiation media for 48 h (DMEM/F12 + 5% FBS + 17 nM insulin, 0.1 μM dexamethasone, 250 μM 3-Isobutyl-1-methylxanthine (IBMX), and 60 μM indomethacin). After 48 h, the differentiation media was replaced by maintenance media (DMEM/F12 + 10% FBS + 17 nM insulin) for a total of 10 days with the maintenance media replaced every 2–3 days.

**Primary mouse adipocyte cAMP assay.** cAMP assay was performed with HTRF dynamic cAMP assay (Cisbio) as previously described with modification for a 24-well plate[2] where adipocytes were incubated in Ham's F12 containing 0.1% BSA and 0.5 mM IBMX plus mouse GIP (Phoenix) in triplicates for 30 min. For muGIPR-Ab treatment, adipocytes were incubated with fresh maintenance media containing muGIPR-Ab overnight, then the media was removed and replaced with Ham's F12 containing 0.1% BSA and 0.5 mM IBMX plus mouse GIP (Phoenix) in triplicates and incubated for 30 min. cAMP concentration of each well was determined using the cAMP dynamic 2 kit (CisBio) following manufacturer's instructions as described above and data expressed as the ratio of 665 nm/620 nm, which is inversely proportional to cAMP concentration where 0 represents maximal cAMP concentration.

**Primary mouse adipocyte lipolysis.** Mature primary adipocytes differentiated as described above treated with mouse GIP (Phoenix) for 2 h compared to basal or after overnight incubation with muGIPR-Ab then treated with 10 nM mouse GIP for 2 h in Krebs-Ringer bicarbonate (KRB) buffer at pH 7.4 with 4% BSA (Sigma) and 5 mM glucose (Sigma). Glycerol released into the medium was determined as lipolytic activity using a fluorometric assay previously described[36] where samples were extracted by adding equal volume of 0.65 N perchloric acid, vortexed, incubated on ice for 10 min, and then neutralized with imidazole-KCl-KOH to precipitate the BSA from solution. Samples were centrifuged for 5 min at 14,000 rpm at 4 °C. Samples were aliquoted in 50 μL dilutions in triplicate to a flat white 96-well microplate, and 50 μL reaction mix (10 mL glycine buffer, 75. U glycerol phosphate dehydrogenase

type I, 15.3 U glycerol kinase, and 115 μL hydrazine hydrate) added to each well. The plate was mixed on a rotor plate for 10 min at 200 rpm then fluorescence measured (excitation: 350 nm, emission: 466 nm) for time 0 measurement. In total, 3 μL 2.5% NAD+ was added to each well and the plate was mixed on a rotor plate for 10 min at 200 rpm then incubated at room temperature for a total incubation time of 45 min. After 45 min incubation, fluorescence was measured (excitation: 350, emission: 466) and time 0 measurement was subtracted from time 45 min measurement, and results were calculated using an interpolated standard curve (sigmoidal, 4PL, X is log concentration) using GraphPad Prism v. 7.02.

**Fatty acid uptake assay in mouse primary adipocytes**. Primary adipocyte fatty acid uptake was measured using the QBT Fatty Acid Uptake Assay Kit (Molecular Devices) as previously described[37] with modification for a 24-well plate. Briefly, adipocytes were incubated with GIP or insulin in DMEM/F12 at 37 °C for 30 min then 2X loading buffer (Molecular Devices) was added per well for a final 1X dilution then the plate was immediately read for kinetic fluorescence (excitation = 485 nm, emission = 515 nm) every 20 s for 1 h. Fatty acid uptake rate was determined as the slope of the line from time 0–5 min (RFU/second).

**GIP-stimulated fatty acid uptake assay in vivo**. Fatty acid uptake in DIO mice was performed as previously described[38] with modifications optimized from internal pilot studies. DIO mice fed HFD for 12 weeks pre-treated with vehicle or muGIPR-Ab (25 mg/kg) for 24-h then were IP dosed with saline or DA-GIP (150 nmol/kg; Phoenix Pharmaceuticals) and simultaneously oral gavaged with 200 μL of 2 μCi $^{14}$C-oleic acid (American Radiolabeled Chemicals) in olive oil (Thermo-Fisher) after a 6-h fast to assess in vivo fatty acid uptake. Blood samples were collected over time by RO bleed, and plasma insulin measured over time by ELISA (Alpco) and plasma radioactivity measured over time by scintillation counting (ThermoFisher). Radioactivity uptake into metabolically relevant tissues was determined by CPM/mg tissue at necropsy (180 min post-dose) by digesting the tissue using Biosol (National Diagnostics) and scintillation counting using Bioscint (National Diagnostics) according to manufacturer's instructions.

For the 6-day LA-Agonist and muGIPR-Ab in vivo fatty acid uptake assay, mice fed HFD for 11 weeks were acclimated to daily saline IP injections for 6 days prior to the study, food intake was measured daily, and mice that shredded their food during acclimation were removed before treatment began, as it made the food intake assessment unreliable. On week 12 HFD feeding, mice were randomized into treatment groups based on body weight and average food intake. Mice were IP injected daily for 6 days with vehicle, muGIPR-Ab (25 mg/kg on day 1, vehicle on days 2–6), or LA-Agonist (37.5 mg/kg/day), and body weight and food intake were measured daily. On day 6, mice were dosed appropriately, fasted for 5 h, RO bled for baseline glucose measurement, then the in vivo FA uptake assay was performed as described above after a 6-h fast.

**Generation of $Gipr^{Adipo-/-}$ mice**. Mice expressing Cre recombinase driven by the adiponectin promoter were generated by Horizon Discovery (now Envigo). An Adipoq-Cre BAC clone was prepared by insertion of Cre cDNA into the $Adipoq$ gene, purified, and the BAC DNA sequenced for confirmation. Embryos were injected with a modified BAC containing the Cre sequence at the Adiponectin gene locus. Specifically, the starting ATG and 222 bp of the Adiponectin gene sequence in the BAC was replaced with Cre sequence. Animals were produced via pronuclear microinjection into single-cell embryos followed by embryo transfer to pseudo-pregnant females. The resulting live births were screened for mutations by PCR and genotyping of pups to identify animals positive for BAC insertion. The resulting F1 pups positive for BAC insertion ($n = 4$) were bred to wild-type C57Bl6/n mice (Taconic), and the resulting pups were genotyped and $Cre$ recombinase expression was measured by RT-PCR, and the founder line with the highest expression was chosen to ship to Charles River for backcrossing to C57Bl6/n mice.

A test mating was performed to confirm adipocyte-specific expression of Adipoq-Cre by mating these Adipoq-Cre mice to R26R mice [purchased from Jackson Laboratories B6.129S4-Gt(ROSA)26Sortm1Sor/J] to generate mice that express lacZ in tissues that express Cre recombinase. Expression was confirmed by staining for β-galactosidase in adipose tissue, pancreas, and reproductive tissues of male and female mice (Supplemental Fig. 3a, b). Adipoq-Cre+ mice did not differ from their wild-type Adipoq-Cre- littermates in their response to HFD body weight gain, fat mass, lean mass, or blood glucose (Supplemental Fig. 3c–f).

Adipoq-Cre+ mice were mated to $Gipr^{fl/fl}$ mice (previously described[2]) to generate mice with heterozygous floxed $Gipr$ gene with or without the Adipoq-Cre transgene. Heterozygous floxed $Gipr$ female mice with the Adipoq-Cre transgene were mated to heterozygous floxed $Gipr$ male mice without the Adipoq-Cre transgene to produce homozygous floxed $Gipr$ ($Gipr^{fl/fl}$) mice with or without the Adipoq-Cre transgene. Subsequent progeny were generated by mating $Gipr^{fl/fl}$ male mice (no Adipoq-Cre) with female homozygous floxed $Gipr$ mice with the Adipoq-Cre transgene ($Gipr^{Adipo-/-}$) to produce mice with adipocyte knockout of $Gipr$ ($Gipr^{Adipo-/-}$) and their wild-type $Gipr^{fl/fl}$ littermates were used as controls for all experiments. Primary adipocytes were isolated and pre-adipocytes were differentiated in vitro as described above. Pancreatic islets were isolated as previously described[2] where mice were euthanized by terminal decapitation and the peritoneal cavity exposed. The Sphincter of Oddi was clamped and 4–6 mL of

cold enzyme buffer (1X Hanks Balanced Salt Solution, 25 mM HEPES, 100 mg/L Dnase I, and 1×Penicillin/Streptomycin with Glutamine)/collagenase (1 mg/mL) was introduced via the bile duct. Following inflation, the pancreas was removed and transferred to a 50 mL falcon tube containing 5 mL enzyme buffer/collagenase on ice. The pancreas was digested at 37 °C for 10–20 min and then the tube was shaken by hand 5–10 times. The digestion was stopped by adding 50 mL cold quenching buffer (enzyme buffer + 10% FBS). The islets were collected by centrifugation at 500 rpm for 2 min. The supernatant was removed, and the islet pellet was resuspended in 50 mL quenching buffer and spun again. Following the two washes, the islet pellet was put on a 3 level histopaque (Sigma) gradient and spun 30 min at 2200 rpm, with the centrifuge brake turned off. Following centrifugation, the purified islets were removed from the middle layer and picked into a fresh culture dish containing RPMI. From there, the islets were picked into 2 mL Eppendorf tubes and frozen at −80 °C in Trizol (Invitrogen) for RNA analysis. RNA was isolated using RNeasy micro kit (Qiagen) and was quantified by RT-PCR as previously described[2] where cDNA was synthesized from equal amounts of RNA using SuperScript III First-Strand Synthesis kit (Invitrogen), and gene expression quantified using PowerUp Sybr Green Reagents (Invitrogen) according to manufacturers' instructions. The following primer pairs were used for $Gipr$ expression normalized to Eukaryotic Elongation Factor 2 ($Eef2$): $Gipr$ F: TTGTG TGGGAGCCAATTACA, $Gipr$ R: ACCCAGGGAATGACGAAAAG, $Eef2$ F: AG CGAGGACAAAGACAAGGA, and $Eef2$ R: GGGATGGTAAGTGGATGGTG.

Male mice were fed HFD for 12 weeks and female mice were fed HFD for 8 weeks during which body weight, fat mass, lean mass, and food intake were measured over time. Mice were then IP injected with vehicle or muGIPR-Ab (25 mg/kg) every 6 days for 48 days for males and 67 days for females. At necropsy, liver was dissected, weighed, and snap frozen in liquid nitrogen. Liver RNA was isolated using RNeasy mini kit (Qiagen) and gene expression quantified using Quantigene Plex Gene Expression Assay (Thermo Fisher) using manufacturer's instructions with 100 ng RNA per reaction and data normalized to $Gapdh$.

**Human adipocyte cAMP assay**. Human adipocytes differentiated in vitro were purchased from Zen-Bio and cAMP assay performed as previously described[2] where adipocytes were incubated in Ham's F12 (Gibco) containing 0.1% BSA (Sigma) and 0.5 mM IBMX plus human GIP (Phoenix Pharmaceuticals) in triplicates for 30 min. cAMP concentration of each well was determined using the cAMP dynamic 2 kit (CisBio) following manufacturer's instructions as described above.

**Mouse endogenous GIPR cell lines cAMP assay**. Mouse Neuro-2a neuro-blastoma cells (ATCC) and rat INS1 832/13 insulinoma cells (EMD Millipore) were used to measure GIP-stimulated cAMP production in a homogeneous time-resolved fluorescence (HTRF) assay (Cisbio,). Neuro-2a cells were cultured in Minimum Essential Media (MEM) w/ Earle's Salts, supplemented with 10% fetal bovine serum (FBS) and 1% penicillin/streptomycin and INS1 cells were cultured in Roswell Park Memorial Institute (RPMI-1640) medium supplemented with 10% FBS, 2 mM L-glutamine, 1 mM sodium pyruvate, 10 mM $N$-2-hydro-xyethylpiperazine-$N$-2-ethane sulfonic acid (HEPES), and 0.05 mM β-mercaptoethanol in a humidified incubator maintained at 37 °C and 5% CO2. Each cell line was plated at 40,000 cells per well and pre-incubated for 24 h with 1 μM of mouse or rat GIP (Phoenix Pharmaceuticals) in assay buffer (0.1% bovine serum albumin in MEM or RPMI-1640 media). The GIP pre-incubation was removed, and cells were washed with phosphate-buffered saline (PBS). The cells were then incubated for 15 min with serially diluted GIP in assay buffer containing 500 μM 3-Isobutyl-1-methylxanthine (IBMX). Cells were then lysed with lysis buffer containing cAMP-d2 and cAMP cryptate (Cisbio) and incubated for 1 h at room temperature before measurement in the Envision plate reader (PerkinElmer). The cAMP levels were expressed as a fluorescence ratio of 665/620 nm. The data points were then fit using a log (agonist) versus response, variable slope (4 parameters) in GraphPad Prism to generate EC$_{50}$ curves.

**I-125 GIP membrane binding**. HEK293T cells overexpressing GIPR were used to determine homologous $^{125}$I-GIP membrane binding. Cells were treated with or without 1 μM DA-GIP (Tocris 6699) in PDL-coated 96-well plates (Corning 354651) and incubated overnight in serum-free DMEM (GIBCO 11965084) in 5% CO$_2$ at 37 °C. The following day, unbound DA-GIP was washed off, and cells were assayed binding using $^{125}$I-labeled human GIP (Perkin Elmer NEX402) in binding buffer (50 mM Hepes buffer, pH 7.2, supplemented with 0.5% BSA) for 2 h in room temperature. After incubation, cells were washed in ice-cold binding buffer and lysed using 200 mM NaOH with 1% SDS for 30 min. Lysate samples were then transferred to Polystyrene NBS plates (Corning 3600) prior to addition of WGA PVT SPA beads (Perkin Elmer RPNQ0001) and incubated overnight in room temperature. The radioactivity was measured as cpm in a TopCount NXT gamma counter (PerkinElmer Life Sciences). Concentration-response curves were fitted to a nonlinear regression (one site—specific binding) in GraphPad Prism (version 7.04, GraphPad Software, La Jolla, CA).

**Rhodamine-GIP membrane binding**. CHOK1 cells stably expressing human GIPR (CHOK1 + GIPR) and CHOK1 cells were cultured in Ham's F12 (Gibco)

supplemented with 10% FBS, 1% penicillin/streptomycin/glutamine (PSG) with or without 5 µg/ml puromycin. Cells were plated at a density of 20,000 cells/well in 96-well plates and cultured for 4 h at 37 °C, 5% $CO_2$ to allow cell attachment prior to treatment with vehicle (culture media) or 100 nM DA-GIP. After 24 h of treatment, cells were washed three times prior to acclimation to assay buffer (F12 + 0.1% BSA) for 1 h at 37 °C, 5% $CO_2$. The cells were then placed on ice for another 15 min prior to treatment with a dose titration of Rhodamine GIP (8 nM to 2 µM, Phoenix Pharmaceutical) in ice-cold F12 + 0.1% BSA. Cells were incubated with Rhodamine-GIP on ice for 60 min and then washed three times with cold PBS and fixed with 4% paraformaldehyde. Cells were then washed and stained with Hoechst 33342 (Thermo Fisher) for nuclei detection. Cells were imaged with Operetta CLS high content imaging system (Perkin Elmer) and Rhodamine-GIP fluorescence was quantitated using the Harmony analysis software (Perkin Elmer). Data represented as relative fluorescence unit (RFU) with background fluorescence (calculated from CHOK1 cells) subtracted.

**GIPR subcellular localization and internalization imaging**. GIPR subcellular localization and internalization were assessed in CHOK1 cells stably expressing human GIPR (CHOK1 + GIPR) or SNAP tagged human GIPR (CHOK1 + SNAP-GIPR) with CHOK1 as the negative control for background determination. Cells were cultured in F12 medium supplemented with 10% FBS, 1% PSG (CHOK1) with 5 µg/mL puromycin (CHOK1 + GIPR), or 0.5 mg/mL geneticin (CHOK1 + SNAP-GIPR). Cells were plated at a density of 20,000 cells/well in 96-well plates and cultured for 4 h at 37 °C, 5% CO2 to allow cell attachment prior to treatment with vehicle (culture media) or DA-GIP (100 nM). After 24 h of treatment, cells were washed three times prior to acclimation to assay buffer (F12 + 0.1% BSA) for 30 min (CHOK1 + SNAP-GIPR) or 1 h (CHOK1 + GIPR) at 37 °C. CHOK1 + SNAP-GIPR cells were incubated with a cell impermeable SNAP-SurfaceAlexa Fluor 647 substrate (S9136S, New England Biolabs) in F12 + 0.1% BSA for 30 min to label all cell surface GIPR and then washed to remove excess unbound labels. After the acclimation/labeling, cells were re-stimulated with GIP (10 nM) and fixed with 4% paraformaldehyde at the indicated timepoint. Cells were then washed 0.3 M glycine and permeabilized with 0.2% triton-X 100. To detect GIPR in CHOK1 + GIPR cells, cells were first blocked with Odyssey blocking buffer (LiCor) for 1 h at room temperature and incubated with a mouse anti-human GIPR monoclonal antibody at 10 µg/mL (MAB8210, R&D Systems) at 4 °C overnight followed by 1 h incubation with Alexa-Fluor 647 conjugated anti-mouse secondary antibody at 4 µg/mL (A32728, Thermo Fisher) for detection. Hoechst 33342 were used for nuclei detection. Images were captured using Operetta CLS high content imaging system (Perkin Elmer) and analyzed by using the Harmony analysis software (Perkin Elmer) to quantify the subcellular GIPR content expressed as relative fluorescence intensity unit (RFU) or internalized GIPR content as spots with background values (calculated from CHOK1) subtracted.

**Statistics and reproducibility**. All values are presented as mean ± SEM. The significance of differences was determined by an unpaired or paired two-tailed Student's $t$ test for comparing two groups as appropriate, one-way ANOVA was performed when there were 3 or more experimental groups with one experimental outcome being measured, or two-way ANOVA was performed whenever an experimental outcome being measured was influenced by two experimental variables followed by multiple comparisons test if applicable. Repeated measures one- or two-way ANOVA was performed if measuring multiple time points over time. Differences were considered significant when $p \le 0.05$. All statistical parameters can be found in the figure legends. Data was analyzed using Graphpad Prism (versions 7.02 or 7.04).

For metabolomics analysis, all statistical analysis was performed by Metabolon, who were blinded to the experimental treatments, and were not altered in any way by Amgen or the authors. Standard statistical analyses were performed in ArrayStudio on log transformed data using Welch's two-sample $t$-test to test whether two unknown means are different from two independent populations with $p \le 0.05$ indicating statistical significance. To correct for multiple comparisons, the $q$-value method for False Discovery Rate was used at a cutoff $q \le 0.2$[33].

The outcomes of all in vitro work presented have been replicated at least twice and all attempts at replication were successful. For in vivo studies, outcomes have been replicated using alternative experimental models as presented here (i.e. GIP-stimulated fatty acid uptake in acute study, chronic study, and in both $Gipr^{\beta Cell-/-}$ and $Gipr^{Adipo-/-}$). Studies using the LA-Agonist have not been replicated more than once because material availability is extremely limited due to cost and degree of difficulty for synthesis. Studies utilizing $Gipr^{Adipo-/-}$ mice have been replicated at least twice and all attempts at replication were successful.

**Reporting summary**. Further information on research design is available in the Nature Research Reporting Summary linked to this article.

## Data availability
The authors declare that all other data supporting the findings of this study are available within the paper and its supplementary information files. The metabolomics data set is available in Supplementary Data 1 file. Source data are provided with this paper.

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

## Acknowledgements

Thank you to Briana Griego (Amgen) for mouse husbandry assistance and Raul Lazaro (Charles River) for necropsy dissection assistance. Thank you to Dr. Kari Wong (Metabolon) for plasma metabolomics data analysis and interpretation. Thank you to Chris De La Torre and Hajime Hiraragi for histological assessment of tissues from R26R × Adipo-Cre+ mice.

## Author contributions

E.A.K., M.M.V., and D.J.L. conceptualized and designed studies. J.F., B.W., and Y.C. designed and produced the LA-Agonist peptide and compound. E.A.K., M.C., G.S, T.H., J.H., L.A., J.L., and H.L. conducted experiments and provided data. E.A.K. and D.J.L. wrote the manuscript.

## Competing interests

All authors are employees of Amgen Inc. and have received Amgen stock.
