## [Peer Review File · Nature Communications]

Reviewers' Comments:

Reviewer #1:

Remarks to the Author:

Referee comments to the manuscript: "Chronic glucose-dependent insulinotropic polypeptide receptor (GIPR) agonism desensitizes adipocyte" by Killion et al,

This is an interesting study directly comparing effects of GIP agonists and antagonists. It is of great interest as it shows that GIPR activity mimic functional GIPR antagonism thereby supporting the recent demonstration of possibly similar effects of GIP agonists and antagonists .

The manuscript is overall well written and the figures presented in an easy readable form.

However, the lack of line numbers makes it inconvenient to referee. More over a range of major and minor concerns should be addressed before acceptance.

One important limitation of the work is that it is performed in mice. It is likely that the mouse and human GIP systems display certain differences, since agonists and antagonists affect the receptors from rodents and humans differentially (several recent publications have demonstrated this). This, of course, needs to be discussed, and at the same time the authors should discuss the important results obtained with GIP receptor antagonists employed in human studies and the preceding in vitro characterization of these antagonists, which are not mentioned with one word. Also in humans there seems to be huge differences between the various responses (fat tissue or beta cell function) from obese and diabetic individuals. It is not clear how these important features are affecting the mouse results .

The long acting GIP agonist is considerably less potent with respect to cAMP . This might influence responses and desensitization. Moreover, the the authors do not at all venture into the molecular mechanisms involved although this could be useful, and do not refer to recent publications describing that GIPR antagonists increase receptor surface expression.

Regarding the EC 50 calculations from FIG 1 d and e , it is not easy to see how these were derived – since there is no maximum effect observed

One interesting question is how soon desensitization develops? So the desensitization is apparently maintained for a long time but how quickly is it induced? In vitro the internalization may be very fast – perhaps it is in vivo as well? But how does endogenous GIP work at all under these conditions? Or perhaps there is no desensitization to endogenous GIP?

p. 2 L. 9-5 bottom are confusing and need better explanation.

p. 3. What is "terminal fat mass"? fat mass at termination? Write that in stead

p 2-3: in the text it is not stated whether all the mice in fig 1 were obese, and or diabetic ?

Fig.1 : it must be admitted that the authors are using very high doses – the internalization of the receptor complex is known to be highly concentration dependent, and whether the internalizations are seen after lesser concentrations is not clear

p. 3. Since these experiments are very complex, the authors need to explain very carefully what is compared to what. Albeit repetitious, it is important to state every time what the therapy was. For instance, don't say combination therapy, be explicit and repeat what the combination consisted of.

P4. Avoid abbreviations; arriving at this in the text the ordinary reader may not remember or deduce what eWAT and SKM is. Define again

p.4. Fig 1I-L there are some interesting differences between the weights/fat content of the various tissues. Would the authors care to discuss these differences?

Fig 1 M: So the insulin response to DA-GIP was preserved after longterm agonism ? how is that possible?or were these acute experiments with LAGIP alone (no preexposure)? This is unclear

p.7 "Adipocytes pre-treated with saline or DA-GIP then stimulated with saline or insulin did not differ in their rates of FA uptake, whereas adipocytes pre-treated with DA-GIP did not respond to subsequent GIP stimulation seen in adipocytes pre-treated with saline (Fig. 2C)." This sentence is incomprehensible to me (and therefore probably also to others)

p. 8 : constitutively internalized" What exactly does that mean?

p. 8 the authors boast about being the first to demonstrate ... certain features about cAMP. While this is tedious to read, it is more interesting to note that the cAMP signal is probably related to lipolysis, whereas the more interesting feature here is FFA + glycerol uptake. So are the authors

that the two processes are occurring at the same time?

Reviewer #2:

Remarks to the Author:

The authors aim to clarify some of the controversy related to the targeting of GIP receptor in prevention and treatment of obesity. This is because GIPR agonists appear to induce similar responses as GIPR antagonists.

In a series of studies using the newly developed GIPR antagonist (anti-GIPR-Ab) and GIPR agonist, at different dosages and in some experiments also in combination with GLP-R agonist (liraglutide), the author show that the similar action of agonist and antagonists is due to similar action in the adipose tissue, where agonism desensitizes GIPR activity, leading to similar function as in antagonism. Overall, the study is important as it does offer a clarification of somewhat unexpected similar actions of agonists & antagonists of GIPR.

Blood-based metabolic signature, based on metabolomics, was also similar for both agonist and antagonist. Somewhat unexpectedly, both agonist and antagonist led to decrease of cAMP, corticosterone and glycerol. However, in this analysis, it is unclear how statistical analyses were performed. It does appear correction for multiple comparisons were not performed. The total dataset includes about 500 metabolites, as can be seen in the supplement (but not clear in the main manuscript).

The supplement also includes pathway annotations of metabolites, and one can see by visual inspection that there are many similarities while there may also be specific differences between the two drugs. To get a global view of metabolic impact of antagonists vs. agonist, it would have been informative if pathway analysis (using e.g. pathway enrichment) would be done.

Similarly for the adipose tissue, since the emphasis of the study is on elucidating shared and specific functionalities for the two different treatment approaches, the pharmacological approach as presented may not go far enough. Since the desensitization of WAT appears to be centrally important in explaining the similar functionality of GIPR agonists and antagonists, it would have been informative to get a genome-wide view of differences, e.g. by acquiring RNA-seq data.

Reviewer #3:

Remarks to the Author:

The paper submitted by Killion and colleagues seeks to understand the effects of chronic stimulation of the GIPR on systemic and adipocyte physiology. The authors hypothesize that chronic stimulation of the GIPR will cause receptor desensitization and effects similar to chronic antagonism. Using a long acting GIP agonist and a GIPR antagonizing Ab they demonstrate comparable effects on body weight in 3 wk treatments, and also reduce plasma glycerol, cAMP, corticosterone and changes in plasma metabolites. Experiments with primary mouse adipocytes suggests that chronic exposure to GIP reduces subsequent cAMP generation, lipolysis, and FA uptake in response to acute stimulation with GIP; this effect is reversible over time. The authors conclude from these data that chronic agonism of the GIPR causes physiologic responses similar to chronic receptor antagonism.

Strengths of the paper:

- The concept incorporates novel aspects regarding incretin receptors. Specifically, do long-acting agonists that provide 24hr exposure have efficacy limited by induction of significant receptor desensitization (by some mechanism).
- The approach utilizes multiple models, including pharmacology, physiology, novel reagents,

metabolomics, and mouse genetics.

- The results of the in vivo and ex vivo experiments are compatible and mutually supportive.
- The finding that GIP has effects on FA uptake that are independent of insulin is novel.

Weaknesses of the paper:

- The authors do not discuss the differential effects of chronic GIPR agonism/antagonism in different tissues. They suggest from in vivo studies that the adrenal gland and adipose tissue is prone to desensitization by chronic exposure to GIP; this is confirmed ex vivo in adipocytes. However the insulin response is not affected, suggesting a different response in β -cells. Moreover, the body weight and food intake effects of GIP-LA suggest that neurons regulating energy balance must be affected similarly to adipocytes. These differential responses are important to understand.
- It is unclear how reducing FFA uptake in adipocytes will produce a meaningful decrease in body weight. Unless the FFAs are redirected towards oxidation in another tissue, it would be expected that they end up stored in another tissue. Some of the data with the GIPR-Ab seems to indicate that blocking FFAs increase liver lipid accumulation (Figure 3f), but this is not a repeated observation (Figure 3o). This concept is not discussed.
- Receptor desensitization is a broad term that can encompass multiple different mechanisms. The data clearly supports a decrease in functional output (FFA uptake, glycerol release) following chronic agonism, but this does not indicate desensitization. Without knowing receptor availability, through binding affinity assays or direct measurement of membrane bound GIPR, it is difficult to identify the mechanism that is responsible for the reduced output. One mechanism could be receptor uncoupling, where the receptor is available for the ligand, but fails to produce the intracellular signaling due to alterations in the G-protein, beta-arrestins, or GRKs. A second mechanism could be reduced receptor availability due to enhanced endocytosis. Very different mechanisms with different outcomes.
- It is somewhat perplexing that GIP treatment of adipocytes causes both lipolysis and FA uptake. This should be commented on.

Technical limitations:

- Essential information is missing on the long-acting GIPR agonist. While it is clear that it possesses GIPR activity (Figure 1a), the specificity is not reported. How much activity occurs in GIPR knockout mice? At the very high dose of 37.5 mg/kg, it is very possible that promiscuous activity at other class B GPCRs could be involved.
- The data in Figure 1 m/n is difficult to qualify. Liraglutide treatment alone reduces GIPR-stimulated insulin secretion, without changing the glucose values. In line with the above comment, this implies understanding the LA-agonist's activity on the GLP-1R is paramount. The effects of the combination treatment (LA-agonist + liraglutide, GIPR-Ab + liraglutide) are impossible to interpret because of the big differences in body weight and presumably insulin sensitivity.
- The modified glucose tolerance test is not a useful measurement here. A single time point for glucose can be very misleading. For example, liraglutide treatment alone produced a substantial amount of body weight loss, which has been reported many times to improve glucose tolerance. This is not reflected in the data in Figure 1n. Moreover, it seems odd that the time point for the glucose value shifts between experiments (Figure 1b = 60 mins, Figure 1n = 80 mins).
- Pg 3, final paragraph, first sentence. This experimental approach does not allow for the determination of metabolic alterations independent of body weight differences. It is unclear what the rationale here is. Why not include controls (no GIP) in these experiments?
- The use of circulating concentrations of cAMP is unusual. Is there validation for plasma concentrations varying based on the activity of a single GPCR ligand?
- Figures 2b-d are missing the untreated control. This is necessary to determine if the basal concentrations of cAMP, corticosterone, and glycerol are reduced, or if the ability for GIP to enhance these parameters are reduced.
- There are controls missing for the data in Figure 3 j-o
- The dose of labelled FFA is not reported. Is this given relative to body weight? For figure 3 p-w, should this be given relative to fat mass?
- Many of the studies seem underpowered. This results in data being reported as not statistically

different, when it appears that it would be if properly powered.

Reviewer #1 (Remarks to the Author):

Referee comments to the manuscript: "Chronic glucose-dependent insulintropic polypeptide receptor (GIPR) agonism desensitizes adipocyte" by Killion et al,

This is an interesting study directly comparing effects of GIP agonists and antagonists. It is of great interest as it shows that GIPR activity mimic functional GIPR antagonism thereby supporting the recent demonstration of possibly similar effects of GIP agonists and antagonists .

The manuscript is overall well written and the figures presented in an easy readable form. However, the lack of line numbers makes it inconvenient to referee. More over a range of major and minor concerns should be addressed before acceptance.

Thank you for your constructive feedback, these comments dramatically improved that manuscript scientifically and stylistically. We apologize for not including line numbers in the first submission and have now done so.

One important limitation of the work is that it is performed in mice. It is likely that the mouse and human GIP systems display certain differences, since agonists and antagonists affect the receptors from rodents and humans differentially (several recent publications have demonstrated this). This, of course, needs to be discussed, and at the same time the authors should discuss the important results obtained with GIP receptor antagonists employed in human studies and the preceding in vitro characterization of these antagonists, which are not mentioned with one word.

We thank the reviewer by bringing to our attention our oversight in mentioning the important work of Rosenkilde et al, especially since the GIP(3-30)NH₂ antagonist has been assessed in humans. GIP(3-30)NH₂ is a well described potent antagonist for the human GIPR, and has been used effectively in human studies to prevent GIP-induced insulin secretion¹ and triglyceride uptake in adipose tissue by altering adipose tissue blood flow². The reviewer is correct in stating this antagonist binds the rodent receptors differentially (10-40 fold lower binding affinity as determined by a radioligand competition assay³, possibly owing to the sequence divergence of the rodent GIP peptide. Despite the effectiveness of the GIP(3-30)NH₂ in human, its poor half-life limits its utility as both a pharmacological agent and a tool compound for in vivo efficacy studies.

Another example of GIP/GIPR differences between primates and rodents is shown by (Pro3)GIP, which is a full agonist on human GIPR but a partial agonist and competitive antagonist on rodent GIPR, and has been used in mice to make claim of the effect of antagonism. The agonism at the human GIPR therefore precludes it utility as an antagonist in human studies⁴. While we agree that we should reference the work with the GIP(3-30)NH₂ in our manuscript (now added to the limitations sections), we interpret these species differences to possibly relate to sequence divergence of the GIP ligand and the GIPR extracellular domain and may not represent an intrinsic difference in GPCR activation and G-protein recruitment. As such, we fully expect our studies in mice using a mouse-specific antagonistic antibody to GIPR to translate to human studies using a human antagonistic antibody (see our previous work with hGIPR-Ab⁵).

Additionally, we also have now conducted the desensitization experiment in human adipocytes and observe identical findings to the mouse adipocytes and have included these data in the new Fig. 5F.

We have now included the following paragraph in the limitations section to highlight the important points raised by the reviewer:

“An important caveat is that these in vivo studies were performed in mice while other GIPR antagonists have been assessed in humans. GIP(3-30)NH₂ is a well described potent antagonist against human GIPR, and has been used effectively in human studies to prevent GIP-induced insulin secretion¹ and triglyceride uptake in adipose tissue by altering adipose tissue blood flow². GIP(3-30)NH₂ binds

rodent GIPR differentially (10-40 fold lower binding affinity as determined by a radioligand competition assay³), possibly owing to the sequence divergence of the rodent and human GIP peptides. Despite the effectiveness of the GIP(3-30)NH₂ in human, its poor half-life limits its utility as both a pharmacological agent and a tool compound for preclinical weight loss studies. We interpret these species differences to possibly relate to sequence divergence of the GIP ligand and the GIPR extracellular domain between rodents and humans and may not represent an intrinsic difference in GPCR activation and G-protein recruitment. Based on our previous work using a human anti-GIPR antibody in efficacy studies in nonhuman primates that had improved efficacy on weight loss compared to muGIPR-Ab in DIO mice⁵, we think the use of a mouse-specific antagonistic antibody to GIPR (muGIPR-Ab) will translate to human studies, particularly since the GIP EC₅₀ for cAMP in mouse primary adipocytes (3.8 nM, Fig. 2M) and in human primary adipocytes (1.3 nM⁵) are similar, and here we have demonstrated identical ligand induced GIPR desensitization in both mouse and human primary adipocytes (Fig. 5A, F)."

Also in humans there seems to be huge differences between the various responses (fat tissue or beta cell function) from obese and diabetic individuals. It is not clear how these important features are affecting the mouse results .

We have now added the following sentence to the limitation section:

"However, differences in responsiveness between obese and diabetic individuals exist that cannot be accounted for by utilizing DIO mice, though as shown in Fig. 3P-W, lean mice do not differ in body weight, adipose tissue or liver weight, or GIP-stimulated FA uptake after treatment with muGIPR-Ab as seen in DIO mice, suggesting inherent differences in GIPR signaling between lean and obese."

The long acting GIP agonist is considerably less potent with respect to cAMP. This might influence responses and desensitization. Moreover, the the authors do not at all venture into the molecular mechanisms involved although this could be useful, and do not refer to recent publications describing that GIPR antagonists increase receptor surface expression.

The molecular mechanisms of desensitization have been previously described by McGraw's group, in which they show constitutive recycling of GIPR and the recycling rate is slower upon ligand addition, leading to reduced cell surface expression^{6,7}. While Rosenkilde's study with the peptidic antagonist GIP(3-30)NH₂ actually restores expression of GIPR thus increases receptor surface expression³, we speculate that the increased surface expression of GIPR with GIP(3-30)NH₂ is functionally inactive owing to the presence of the peptide antagonist and thus renders the GIPR non-functional despite the increase in surface receptors. Because our manuscript is primarily focused on the established desensitization with GIPR agonists using primary adipocytes, rather than immortalized cell lines, we have referenced the McGraw's paper as a possible molecule mechanism.

Additionally, at the suggestion of the reviewer, we have now tested this published mechanism and assessed cell surface disappearance of GIPR using both imaging and radioligand binding experiments as supportive data of the McGraw and Rosenkilde findings, and we now included these data and can confirm long term exposure of GIPR to chronic GIPR agonism reduces surface expression of GIPR (Fig. 5I-M, Supplemental Fig. 3).

Regarding the potency differences between LA-GIPR Agonist and DA-GIP, we have now tested mouse GIP (EC₅₀ = 4.6 pM), DA-GIP (EC₅₀ = 18.3 pM), the LA-Agonist (EC₅₀ = 505 pM), and the peptide only portion of the LA-Agonist (EC₅₀ = 4.1 pM). We hypothesize the potency differences in vitro may be due to the antibody portion of the LA-Agonist conjugate interfering with optimal receptor interaction in vitro. In vivo as shown in Fig. 1C, with both DA-GIP and LA-Agonist dosed at 50 nmol/kg, the LA-Agonist (purple) shows increased insulin secretion than DA-GIP (blue) likely due to its increased half-life.

Regarding the EC 50 calculations from FIG 1 d and e , it is not easy to see how these were derived – since there is no maximum effect observed

We were able to derive the EC50s using GraphPad Prism software and it becomes much easier to see how they were derived by using the means of the data than the individual values We have updated the graphs in Figure 2D, E to reflect the means. We do agree that it would have been ideal to have a higher dose for the blood glucose graph while the insulin secretion graph is easier to see that the effect is maxed out. To be very conservative in selecting the dose to assess maximal efficacy of the LA-Agonist, we chose to cover 10x the EC₅₀ for glucose (328 nM) rather than insulin (212 nM), which would be less conservative.

One interesting question is how soon desensitization develops? So the desensitization is apparently maintained for a long time but how quickly is it induced? In vitro the internalization may be very fast – perhaps it is in vivo as well? But how does endogenous GIP work at all under these conditions? Or perhaps there is no desensitization to endogenous GIP?

We agree that the rate of desensitization is important to establish as previous experiments were all incubated with 1 uM DA-GIP for 24 hours. We now include new data in which we determined the time course for DA-GIP desensitization of the receptor with respect to cAMP generation: reduced responsiveness was observed after 1 and 4 hours pre-treatment but full desensitization was maximally seen at 24 hours (Fig. 5C). Along these lines, we also assessed the dose-response of DA-GIP pre-treatment for 24 hours and saw that even 10 nM DA-GIP blunted the responsiveness to 100 nM GIP re-stimulation with maximal inhibition seen at 100 nM and 1 uM DA-GIP (Fig. 5D).

On the rate of receptor internalization³, it takes about 30 minutes for maximal internalization. However, we show desensitization requires a lot longer in the presence of the GIP. Our experiments are conducted using DA-GIP for the desensitization experiments to ensure the molecule remains intact for the incubation period but in fact conducted the actual stimulation experiment with native GIP, so we fully expect desensitization in vivo to endogenous GIP.

p. 2 L. 9-5 bottom are confusing and need better explanation.
We have added a better explanation.

p. 3. What is “terminal fat mass”? fat mass at termination? Write that in stead
We have corrected the description.

p 2-3: in the text it is not stated whether all the mice in fig 1 were obese, and or diabetic ?
We have added this clarification.

Fig.1 : it must be admitted that the authors are using very high doses – the internalization of the receptor complex is known to be highly concentration dependent, and whether the internalizations are seen after lesser concentrations is not clear

We agree that in our incubation studies we were using 1 μM DA-GIP without previously optimizing the doses, we have now added additional data showing a dose response and added these data to Fig. 5D. However, it was our intention to recapitulate the concentrations achieved with pharmacological concentrations using GIPR agonist therapies (much higher than endogenous levels).

p. 3. Since these experiments are very complex, the authors need to explain very carefully what is compared to what. Albeit repetitious, it is important to state every time what the therapy was. For instance, don't say combination therapy, be explicit and repeat what the combination consisted of.

We have now carefully stated the experiments.

P4. Avoid abbreviations; arriving at this in the text the ordinary reader may not remember or deduce what eWAT and SKM is. Define again

We understand this and have made sure to define each abbreviation twice in the text, but we do need to utilize abbreviations for the sake of word count.

p.4. Fig 1I-L there are some interesting differences between the weights/fat content of the various tissues. Would the authors care to discuss these differences?

We have added a better explanation.

Fig 1 M: So the insulin response to DA-GIP was preserved after longterm agonism ? how is that possible?or were these acute experiments with LAGIP alone (no preexposure)?

We thank the reviewer for picking up on this very important observation. Yes, we did in fact observe that after long term agonism the insulin secretion to DA-GIP was maintained. We now speculate this might be related to the expression of the GIPR in pancreatic beta cells compared to other tissues and have now included expression levels in Supplemental Figure 4. Indeed, we have extended our data and conducted the desensitization experiments in different cells lines that express GIPR: adipocytes (mouse and human), mouse Neuro-2A neuroblastoma cells, and rat islet INS1 832/13 cells (Fig. 5A, F, G, H). In both adipocytes and Neuro2A cells, DA-GIP leads to desensitization, whereas in the INS1 cells, the cAMP levels remain high even after 24 hours of incubation with DA-GIP demonstrating a lack of desensitization. Therefore, it is possible that cell types with lower expression levels are more amenable to desensitization and we have added the following paragraph to the discussion:

“From the data demonstrating similarities in GIPR ligand-mediated desensitization in primary mouse adipocytes, primary human adipocytes, and mouse Neuro2a neuroblastoma cell lines, we hypothesize that pharmacological GIPR agonism does not only desensitize GIPR in adipocytes but may be a feature of cell types with relatively low levels of GIPR expression. Indeed, both the LA-Agonist and muGIPR-Ab display significantly reduced cumulative food intake after six days of treatment (Fig. 5N), indicating that ligand-mediated desensitization in neuronal cell types may also occur as reflected by desensitization in mouse Neuro-2a neuroblastoma cells (Fig. 5G). This is also supported by recent work from two different groups demonstrating that central administration of either GIP⁸ and of an anti-GIPR antibody⁹ both lead to a reduction in food intake, which we hypothesize can be effectively explained by desensitization of GIPR in neuronal cells by GIP leading to the same phenotype as an anti-GIPR antibody. Conversely, in a pancreatic islet cell line, rat INS1 832/13, cells appear to not desensitize, but rather display high basal levels of cAMP after chronic GIPR agonism (Fig. 5H), suggesting that GIPR continues to signal in these cells. Similarly, mice treated with LA-Agonist for 21 days were able to respond to DA-GIP stimulated insulin secretion 4 hours after the last LA-Agonist dose (Fig. 1M), which suggests that GIPR in pancreatic β -cells does not desensitize in a manner like adipocytes or neuronal cells. We hypothesize that cell type differences in desensitization can be explained by differences in GIPR expression where pancreatic islets have dramatically higher levels of *Gipr* expression compared to adipose tissue and brain sections (Supplemental Fig. 4). However, further work is needed to understand whether these cell type differences are due to differences in expression level or other components of cell signaling. We now conclude that anti-GIPR antibodies do not require GIPR activity in pancreatic β -cells for efficacy⁵, but rather partially depend on GIPR activity in adipocytes and further work is needed to find additional tissues or cell types for which GIPR therapies are dependent on, such as neuronal subpopulations as investigated by recent literature^{8,9}.”

p.7 “Adipocytes pre-treated with saline or DA-GIP then stimulated with saline or insulin did not differ in their rates of FA uptake, whereas adipocytes pre-treated with DA-GIP did not respond to subsequent GIP stimulation seen in adipocytes pre-treated with saline (Fig. 2C).” This sentence is incomprehensible to me (and therefore probably also to others)

We have rewritten.

p. 8 : constitutively internalized” What exactly does that mean?

We have rewritten – “it was shown that GIPR is constitutively internalized, i.e. receptor internalization even in the absence of GIP stimulation, which is an atypical characteristic of most GPCRs.”

p. 8 the authors boast about being the first to demonstrate ... certain features about cAMP. While this is tedious to read, it is more interesting to note that the cAMP signal is probably related to lipolysis, whereas the more interesting feature here is FFA + glycerol uptake. So are the authors that the two processes are occurring at the same time?

In vivo, physiological GIP concentrations likely do not act to promote lipolysis in vivo since the presence of high GIP will also stimulate insulin secretion, and insulin is a potent inhibitor of GIP-stimulated lipolysis¹⁰. However, the mechanism of GIP-stimulated adipocyte lipolysis, rather than triglyceride storage, could potentially be an anti-obesity mechanism for pharmacological doses of GIPR agonists to overcome the anti-lipolytic effects of insulin as recently presented by scientists from Eli Lilly¹¹. It is also important to note that many experiments examining GIP’s direct effects on adipocytes in vitro use non-physiological concentrations of GIP to elicit effects and likely may not reflect in vivo physiological mechanisms; however, these experiments are important to consider because they may reflect pharmacological levels of GIP observed with GIPR agonists in vivo.

We have now added the following paragraph to the discussion:

“We and others have shown that GIP stimulates in vitro lipolysis in the absence of insulin¹²; however, physiological GIP concentrations likely do not act to promote lipolysis in vivo since the presence of high GIP will also stimulate insulin secretion, and insulin is a potent inhibitor of GIP-stimulated lipolysis¹⁰. Similar to insulin, GIP has also stimulates FA uptake and re-esterification in rat adipose tissue^{13,14} and in mouse primary adipocytes ex vivo^{12,15}. However, the mechanism of GIP-stimulated adipocyte lipolysis, rather than triglyceride storage, could potentially be an anti-obesity mechanism for pharmacological doses of GIPR agonists to overcome the anti-lipolytic effects of insulin. On the contrary, here we have demonstrated that in fact chronic GIPR agonism indeed decreases plasma glycerol, a marker of adipocyte lipolysis, as well as inhibits adipose tissue FA uptake, like a GIPR antagonist. Both acutely and chronically, the inhibition of adipose tissue FA uptake leads to an increase in liver FA uptake, and in chronic studies, results in reduced liver weight and increased liver FA oxidation genes, suggesting a redistribution of FA from storage in adipose tissue to oxidation in other tissues.”

Reviewer #2 (Remarks to the Author):

The authors aim to clarify some of the controversy related to the targeting of GIP receptor in prevention and treatment of obesity. This is because GIPR agonists appear to induce similar responses as GIPR antagonists.

In a series of studies using the newly developed GIPR antagonist (anti-GIPR-Ab) and GIPR agonist, at different dosages and in some experiments also in combination with GLP-R agonist (liraglutide), the author show that the similar action of agonist and antagonists is due to similar action in the adipose tissue, where agonism desensitizes GIPR activity, leading to similar function as in antagonism. Overall, the study is important as it does offer a clarification of somewhat unexpected similar actions of agonists & antagonists of GIPR.

Blood-based metabolic signature, based on metabolomics, was also similar for both agonist and antagonist. Somewhat unexpectedly, both agonist and antagonist led to decrease of cAMP, corticosterone and glycerol. However, in this analysis, it is unclear how statistical analyses were performed. It does appear correction for multiple comparisons were not performed. The total dataset includes about 500 metabolites, as can be seen in the supplement (but not clear in the main manuscript).

We apologize for the oversight in previously describing how the metabolomics statistical analysis were performed. The previous version of the manuscript was written as a short report and in an effort to be as concise as possible, we did not fully explain Metabolon's statistical analysis. We have now highlighted this in the methods section for both metabolomics analysis and statistical analysis as well as highlighted it in the figure legend. All statistical analysis was performed by Metabolon, who were blinded to the experimental treatments, and were not altered in any way by Amgen or the authors. Standard statistical analyses were performed in ArrayStudio on log transformed data using Welch's two-sample t-test to test whether two unknown means are different from two independent populations with $p < 0.05$ indicating statistical significance. To correct for multiple comparisons and control for type I error, the q-value method for False Discovery Rate was used at a cutoff $q < 0.2$ ¹⁶.

To address your point and make it clear that the total number of metabolites detect is $n = 671$, we have now illustrated the statistically significant changes observed between each comparison were $p < 0.05$ and $q < 0.2$ for vehicle vs muGIPR-Ab (Fig. 2A), vehicle vs LA-Agonist (Fig. 2B), and muGIPR-Ab vs. LA-Agonist (Fig. 2C).

The supplement also includes pathway annotations of metabolites, and one can see by visual inspection that there are many similarities while there may also be specific differences between the two drugs. To get a global view of metabolic impact of antagonists vs. agonist, it would have been informative if pathway analysis (using e.g. pathway enrichment) would be done.

When comparing samples from mice treated with muGIPR-Ab vs. LA-Agonist, there are a handful of metabolites with $p < 0.05$; however, all of the q-values > 0.95 , which indicates that there is a 95% chance that these are false discoveries and would not meet the criteria for metabolome wide significance of $p < 0.05$ and $q < 0.2$ (Fig. 2C), so there are not any statistically significant metabolites to compare between muGIPR-Ab vs. LA-Agonist for enriched pathways. We used Metabolon's Metabolync software to perform pathway enrichment comparing vehicle vs. muGIPR-Ab and vehicle vs. LA-Agonist (Fig. 2E), and pathways found to be enriched in muGIPR-Ab compared to vehicle were also found to be enriched in LA-Agonist compared to vehicle with the highest enrichment for both treatments being amino sugar metabolism; fatty acid metabolism (acyl choline); and fructose, mannose, and galactose metabolism.

Similarly for the adipose tissue, since the emphasis of the study is on elucidating shared and specific functionalities for the two different treatment approaches, the pharmacological approach as presented may not go far enough. Since the desensitization of WAT appears to be centrally important in explaining the similar functionality of GIPR agonists and antagonists, it would have been informative to get a genome-wide view of differences, e.g. by acquiring RNA-seq data.

We agree that RNA-Seq data would be very exciting and highly informative. Unfortunately, we do not have support at this time for RNA-Seq analysis. We feel that the functional *in vivo* assays that we have developed (FA uptake and cumulative food intake compared head-to-head in Fig. 5N-X) demonstrate both desensitization of GIPR in both adipocytes and neuronal cells. Additionally, we have now included data to show that mice with adipocyte GIPR knockout only partially respond to the effect of the anti-GIPR antibody (Fig. 4M-P), which demonstrates a role for GIPR activity in adipocytes, but also leaves room to further explore other tissues and cell types, such as the brain. We utilized mouse Neuro-2a neuroblastoma cells to demonstrate that the same ligand-mediated receptor desensitization seen in adipocytes also occurs in neuronal cells (Fig. 5G), which may explain why both chronic GIPR agonism and antagonism both lead to reduced food intake (Fig. 5N).

Reviewer #3 (Remarks to the Author):

The paper submitted by Killion and colleagues seeks to understand the effects of chronic stimulation of the GIPR on systemic and adipocyte physiology. The authors hypothesize that chronic stimulation of the GIPR will cause receptor desensitization and effects similar to chronic antagonism. Using a long acting GIP agonist and a GIPR antagonizing Ab they demonstrate comparable effects on body weight in 3 wk treatments, and also reduce plasma glycerol, cAMP, corticosterone and changes in plasma metabolites. Experiments with primary mouse adipocytes suggests that chronic exposure to GIP reduces subsequent cAMP generation, lipolysis, and FA uptake in response to acute stimulation with GIP; this effect is reversible over time. The authors conclude from these data that chronic agonism of the GIPR causes physiologic responses similar to chronic receptor antagonism.

Strengths of the paper:

- The concept incorporates novel aspects regarding incretin receptors. Specifically, do long-acting agonists that provide 24hr exposure have efficacy limited by induction of significant receptor desensitization (by some mechanism).
- The approach utilizes multiple models, including pharmacology, physiology, novel reagents, metabolomics, and mouse genetics.
- The results of the *in vivo* and *ex vivo* experiments are compatible and mutually supportive.
- The finding that GIP has effects on FA uptake that are independent of insulin is novel.

Weaknesses of the paper:

- The authors do not discuss the differential effects of chronic GIPR agonism/antagonism in different tissues. They suggest from *in vivo* studies that the adrenal gland and adipose tissue is prone to desensitization by chronic exposure to GIP; this is confirmed *ex vivo* in adipocytes. However the insulin response is not affected, suggesting a different response in β -cells. Moreover, the body weight and food intake effects of GIP-LA suggest that neurons regulating energy balance must be affected similarly to adipocytes. These differential responses are important to understand.

We appreciate the interest from the reviewer on the effects of antagonism/ agonism in different tissues, especially since we observe desensitization in adipocytes and adrenals but not in pancreatic beta cells (insulin response). We thank the reviewer for picking up on this very important observation. Yes, we did in fact observe that after long term agonism the insulin secretion to DA-GIP was maintained. We now speculate this might be related to the expression of the GIPR in pancreatic beta cells compared to other tissues and have now included expression levels in Supplemental Figure 4. Indeed, we have extended our data and conducted the desensitization experiments in different cell lines that express GIPR: adipocytes (mouse and human), mouse Neuro-2A neuroblastoma cells, and rat islet INS1 832/13 cells (Fig. 5A, F, G, H). In both adipocytes and Neuro2A cells, DA-GIP leads to desensitization, whereas in the INS1 cells, the cAMP levels remain high even after 24 hours of incubation with DA-GIP demonstrating a lack of desensitization. Therefore, it is possible that cell types with lower expression levels are more amenable to desensitization and we have added the following paragraph to the discussion:

“From the data demonstrating similarities in GIPR ligand-mediated desensitization in primary mouse adipocytes, primary human adipocytes, and mouse Neuro2a neuroblastoma cell lines, we hypothesize that pharmacological GIPR agonism does not only desensitize GIPR in adipocytes but may be a feature of cell types with relatively low levels of GIPR expression. Indeed, both the LA-Agonist and muGIPR-Ab display significantly reduced cumulative food intake after six days of treatment (Fig. 5O), indicating that ligand-mediated desensitization in neuronal cell types may also occur as reflected by desensitization in mouse Neuro-2a neuroblastoma cells (Fig. 5G). This is also supported by recent work from two different groups demonstrating that central administration of either GIP⁸ and of an anti-GIPR antibody⁹ both lead to a reduction in food intake, which we hypothesize can be effectively explained by desensitization of GIPR in neuronal cells by GIP leading to the same phenotype as an anti-GIPR antibody. Conversely, in a pancreatic islet cell line, rat INS1 832/13, cells appear to not desensitize, but rather display high basal levels of cAMP after chronic GIPR agonism (Fig. 5H), suggesting that GIPR continues to signal in these cells. Similarly, mice treated with LA-Agonist for 21 days were able to respond to DA-GIP stimulated insulin secretion 4 hours after the last LA-Agonist dose (Fig. 1M), which suggests that GIPR in pancreatic β -cells does not desensitize in a manner like adipocytes or neuronal cells. We hypothesize that cell type differences in desensitization can be explained by differences in GIPR expression where pancreatic islets have dramatically higher levels of *Gipr* expression compared to adipose tissue and brain sections (Supplemental Fig. 4). However, further work is needed to understand whether these cell type differences are due to differences in expression level or other components of cell signaling. We now conclude that anti-GIPR antibodies do not require GIPR activity in pancreatic β -cells for efficacy⁵, but rather partially depend on GIPR activity in adipocytes and further work is needed to find additional tissues or cell types for which GIPR therapies are dependent on, such as neuronal subpopulations as investigated by recent literature^{8,9}.”

Additionally, we are now able to include data that mice with adipocyte GIPR knockout only partially respond to the effect of the anti-GIPR antibody (Fig. 4M-P), which demonstrates a role for GIPR activity in adipocytes, but also leaves room to further explore other tissues and cell types, such as the brain. We utilized mouse Neuro-2a neuroblastoma cells to demonstrate that the same ligand-mediated receptor desensitization seen in adipocytes also occurs in neuronal cells (Fig. 5G), which may explain why both chronic GIPR agonism and antagonism both lead to reduced food intake (Fig. 5N).

- It is unclear how reducing FFA uptake in adipocytes will produce a meaningful decrease in body weight. Unless the FFAs are redirected towards oxidation in another tissue, it would be expected that they end up stored in another tissue. Some of the data with the GIPR-Ab seems to indicate that blocking FFAs increase liver lipid accumulation (Figure 3f), but this is not a repeated observation (Figure 3o). This concept is not discussed.

We agree that the redirection of FFAs to the liver may indeed represent a mechanism for the weight loss. Indeed, we observed a redirection of FFA to the liver in our radiolabeled uptake studies following a single dose of the GIPR antagonist (Fig. 3F) and in a chronic study (Fig. 3W). In chronic studies, we consistently find a decrease in liver weight (Fig. 3S, Fig. 4Q). To investigate this further at the request of the reviewer, we looked at gene expression of hepatic fatty acid oxidation and lipogenesis genes in the livers of GIPR antagonist dosed mice. We found that liver from *Gipr^{fl/fl}* mice treated with muGIPR-Ab had a modest, but significant increase in many genes associated with hepatic FA oxidation, including *Ppara*, *Acaa2*, *Acadm*, and *Cpt2* (Fig. 4R), without alteration in genes associated with hepatic lipogenesis (Fig. 4S), suggesting that in mice chronically treated with muGIPR-Ab, the reduced liver weight is associated with increased hepatic FA uptake and increased hepatic FA oxidation and is dependent on adipocyte GIPR activity. We have added the new data and the following paragraph in the discussion:

“Here we have demonstrated that in fact chronic GIPR agonism indeed decreases plasma glycerol, a marker of adipocyte lipolysis, as well as inhibits adipose tissue FA uptake, like a GIPR antagonist. For both conditions acutely and chronically, the inhibition of adipose tissue FA uptake leads to an increase in liver FA uptake, and in chronic studies, results in reduced liver weight and increased liver FA oxidation genes, suggesting a redistribution of FA from storage in adipose tissue to oxidation in other tissues.”

- Receptor desensitization is a broad term that can encompass multiple different mechanisms. The data clearly supports a decrease in functional output (FFA uptake, glycerol release) following chronic agonism, but this does not indicate desensitization. Without knowing receptor availability, through binding affinity assays or direct measurement of membrane bound GIPR, it is difficult identify the mechanism that is responsible the reduced output. One mechanism could be receptor uncoupling, where the receptor is available for the ligand, but fails to produce the intracellular signaling due to alterations in the G-protein, beta-arrestins, or GRKs. A second mechanism could be reduced receptor availability due to enhanced endocytosis. Very different mechanisms with different outcomes. The molecular mechanisms have been extensively described by McGraw's group, that is, receptor recycling and reduced cell surface expression^{6,7}. We have independently confirmed the mechanism of receptor desensitization in a set of experiments looking at the subcellular localization of GIPR in a recombinant cell line overexpressing GIPR. Firstly, we looked at both I-125 GIP binding and Rhodamine-labeled GIP binding to cells that had been pre-treated with or without DA-GIP and observed less binding of GIP to the pre-treated cells suggesting a paucity in cell surface GIPR (Fig. 5I and Supplemental Fig. 3a). We followed up on this finding by staining for the localization of GIPR. Again, we saw that in cells that had been pre-treated with DA-GIP, there was loss of cell surface GIPR after 24 hours (Fig. 5J, K and Supplemental Fig. 3A, B). From these data we conclude that loss of cell surface expression is most likely the cause of the reduced response to GIP. Subsequent GIP stimulation brings GIPR internalization in cells without DA-GIP pre-treatment down to the same level as cells pre-treated with DA-GIP (Fig. 5 L, M).

- It is somewhat perplexing that GIP treatment of adipocytes causes both lipolysis and FA uptake. This should be commented on.

In vivo, physiological GIP concentrations likely do not act to promote lipolysis in vivo since the presence of high GIP will also stimulate insulin secretion, and insulin is a potent inhibitor of GIP-stimulated lipolysis¹⁰. However, the mechanism of GIP-stimulated adipocyte lipolysis, rather than triglyceride storage, could potentially be an anti-obesity mechanism for pharmacological doses of GIPR agonists to overcome the anti-lipolytic effects of insulin as recently presented by scientists from Eli Lilly¹¹. It is also important to note that many experiments examining GIP's direct effects on adipocytes in vitro use non-physiological concentrations of GIP to elicit effects and likely may not reflect in vivo physiological mechanisms; however, these experiments are important to consider because they may reflect pharmacological levels of GIP observed with GIPR agonists in vivo.

We have now added the following paragraph to the discussion:

“We and others have shown that GIP stimulates *in vitro* lipolysis in the absence of insulin¹²; however, physiological GIP concentrations likely do not act to promote lipolysis *in vivo* since the presence of high GIP will also stimulate insulin secretion, and insulin is a potent inhibitor of GIP-stimulated lipolysis¹⁰. Similar to insulin, GIP has also stimulates FA uptake and re-esterification in rat adipose tissue^{13,14} and in mouse primary adipocytes *ex vivo*^{12,15}. However, the mechanism of GIP-stimulated adipocyte lipolysis, rather than triglyceride storage, could potentially be an anti-obesity mechanism for pharmacological doses of GIPR agonists to overcome the anti-lipolytic effects of insulin. On the contrary, here we have demonstrated that in fact chronic GIPR agonism indeed decreases plasma glycerol, a marker of adipocyte lipolysis, as well as inhibits adipose tissue FA uptake, like a GIPR antagonist. Both acutely and chronically, the inhibition of adipose tissue FA uptake leads to an increase in liver FA uptake, and in chronic studies, results in reduced liver weight and increased liver FA oxidation genes, suggesting a redistribution of FA from storage in adipose tissue to oxidation in other tissues.”

Technical limitations:

- Essential information is missing on the long-acting GIPR agonist. While it is clear that it possesses GIPR activity (Figure 1a), the specificity is not reported. How much activity occurs in GIPR knockout mice? At the very high dose of 37.5 mg/kg, it is very possible that promiscuous activity at other class B GCPRs could be involved.

We have now further characterized the long-acting GIPR agonist to show its selectivity *in vitro* for GIPR over GLP-1 receptor and glucagon receptor, as well as assessing very high concentrations of both GIP and the long-acting GIPR agonist peptide in mice with beta-cell GIPR knockout for insulin secretion and demonstrated that neither results in GLP-1R-mediated insulin secretion at very high concentrations (Supplemental Figure 1).

- The data in Figure 1 m/n is difficult to qualify. Liraglutide treatment alone reduces GIPR-stimulated insulin secretion, without changing the glucose values. In line with the above comment, this implies understanding the LA-agonists activity on the GLP-1R is paramount. The effects of the combination treatment (LA-agonist + liraglutide, GIPR-Ab + liraglutide) are impossible to interpret because of the big differences in body weight and presumably insulin sensitivity.

We agree with the point raised by the reviewer that the liraglutide alone reduces insulin secretion and likely represents a difference in whole body sensitivity concordant with the dramatic weight loss. We are now confident that the LA-GIPR agonist is not active on the GLP-1R, both *in vitro* (Supplemental Fig. 1B, E). We believe the key piece of data in this figure is the fact that the DA-GIP was still able to induce insulin secretion in the LA-Agonist group whereas it was unable to in the muGIPR-Ab group. We agree that the combination groups are very difficult to interpret because of the dual agents and the strong effects on weight loss but feel it would be incorrect to just show the vehicle, LA-agonist and muGIPR-Ab groups, despite the fact it would greatly simplify the key message.

- The modified glucose tolerance test is not a useful measurement here. A single time point for glucose can be very misleading. For example, liraglutide treatment alone produced a substantial amount of body weight loss, which has been reported many times to improve glucose tolerance. This is not reflected in the data in Figure 1n. Moreover, it seems odd that the time point for the glucose value shifts between experiments (Figure 1b = 60 mins, Figure 1n = 80 mins).

We agree that the GIP-induced changes on glucose measures are very difficult to interpret and in fact could be misleading. We have previously demonstrated that in DIO mice the DA-GIP induced effects are

most profound at 15 minutes for insulin secretion and 60 minutes for glucose changes⁵. We used this experimental design for Fig. 2B, C, but because there were so many mice in the study in Figure 1N, the technician was unable to bleed the first mouse again within a 60 minute timeframe so was only able to start rebleeding after 80 mins. We feel the key piece of data is the insulin secretion, which is a direct measurement of GIPR activity, whereas the differences in blood glucose over time is not only reflective of insulin secretion but also peripheral glucose sensitivity.

- Pg 3, final paragraph, first sentence. This experimental approach does not allow for the determination of metabolic alterations independent of body weight differences. It is unclear what the rationale here is. Our intention was to be able to directly compare the differences in the agonist treated animals compared to the antagonist treated animals since there was no difference in body weight between the groups. We acknowledge there was a difference in body weight between these two groups versus the vehicle. We have modified the sentence to improve clarity of our rationale. – “To determine the metabolic alterations of GIPR agonism compared to antagonism independent of body weight differences, i.e. the LA-Agonist alone and muGIPR-Ab alone both had a modest effect preventing body weight gain while the combination with liraglutide yielded dramatic body weight loss, we utilized untargeted metabolomics analysis of plasma samples from mice chronically treated with vehicle, LA-Agonist, or muGIPR-Ab collected 80 minutes after the glucose + DA-GIP challenge (shown in Fig. 1G-N).”

- The use of circulating concentrations of cAMP is unusual. Is there validation for plasma concentrations varying based on the activity of a single GPCR ligand?

We agree that presenting circulating levels of cAMP is unusual, however since this was unbiased metabolomics we are confident in the validity of the data and this is increased by the observation that treatment groups were different to vehicle. Of note plasma cAMP is a standard clinical laboratory test to assess PTH receptor activity, <https://www.labcorp.com/tests/004984/cyclic-amp-plasma>

- Figures 2b-d are missing the untreated control. This is necessary to determine if the basal concentrations of cAMP, corticosterone, and glycerol are reduced, or if the ability for GIP to enhance these parameters are reduced.

We agree that ideally, we would have had an additional vehicle group in the chronic treatment study that should not have received DA-GIP at the end of the study as an additional control. We do feel however that the study is adequately controlled with the presence of the vehicle group. To improve confidence in the findings in the chronic study we conducted follow-up studies using a no-DA-GIP control group in Fig. 2I, J, and we demonstrate that the effects of DA-GIP are clear in increasing the two metabolites corticosterone and glycerol, thereby providing the appropriate controlled experiment for the prior chronic study. We cannot repeat the chronic study as the long acting agonist was very difficult and expensive to generate. The subsequent Fig. 2K, L demonstrate that the GIPR antagonist prevents the GIP-induced increase for these 3 parameters.

- There are controls missing for the data in Figure 3 j-o

The reviewer refers to a non-DA-GIP treated group in new Fig. 3J-O. However, we show in the previous experiment (Fig. 3C-I) that the assay window is already established and demonstrate the effectiveness of the control in Fig. 3C, D. Since we have generated a robust response of the DA-GIP, we chose to omit it for subsequent experiments because bleeding the animals in 30 min intervals only allows for n = 30 animals/experiment.

- The dose of labelled FFA is not reported. Is this given relative to body weight? For figure 3 p-w, should this be given relative to fat mass?

The previous version of the manuscript was written as a short report and in an effort to be as concise as possible, we did not fully explain the methods but rather just cited the published protocol that we used¹⁷. We have now clarified in the methods that each mouse received 200 μ L of 2 μ Ci ¹⁴C-oleic acid in olive oil.

- Many of the studies seem underpowered. This results in data being reported as not statistically different, when it appears that it would be if properly powered.
We typically use n = 8 animals per group and have had no issues previously with establishing a treatment effect.

References

- 1 Gasbjerg, L. S. *et al.* GIP(3-30)NH2 is an efficacious GIP receptor antagonist in humans: a randomised, double-blinded, placebo-controlled, crossover study. *Diabetologia* **61**, 413-423, doi:10.1007/s00125-017-4447-4 (2018).
- 2 Asmar, M. *et al.* The Gluco- and Liporegulatory and Vasodilatory Effects of Glucose-Dependent Insulinotropic Polypeptide (GIP) Are Abolished by an Antagonist of the Human GIP Receptor. *Diabetes* **66**, 2363-2371, doi:10.2337/db17-0480 (2017).
- 3 Gabe, M. B. N. *et al.* Human GIP(3-30)NH2 inhibits G protein-dependent as well as G protein-independent signaling and is selective for the GIP receptor with high-affinity binding to primate but not rodent GIP receptors. *Biochem Pharmacol* **150**, 97-107, doi:10.1016/j.bcp.2018.01.040 (2018).
- 4 Sparre-Ulrich, A. H. *et al.* Species-specific action of (Pro3)GIP - a full agonist at human GIP receptors, but a partial agonist and competitive antagonist at rat and mouse GIP receptors. *Br J Pharmacol* **173**, 27-38, doi:10.1111/bph.13323 (2016).
- 5 Killion, E. A. *et al.* Anti-obesity effects of GIPR antagonists alone and in combination with GLP-1R agonists in preclinical models. *Sci Transl Med* **10**, doi:10.1126/scitranslmed.aat3392 (2018).
- 6 Abdullah, N., Beg, M., Soares, D., Dittman, J. S. & McGraw, T. E. Downregulation of a GPCR by beta-Arrestin2-Mediated Switch from an Endosomal to a TGN Recycling Pathway. *Cell Rep* **17**, 2966-2978, doi:10.1016/j.celrep.2016.11.050 (2016).
- 7 Mohammad, S. *et al.* A naturally occurring GIP receptor variant undergoes enhanced agonist-induced desensitization, which impairs GIP control of adipose insulin sensitivity. *Molecular and cellular biology* **34**, 3618-3629, doi:10.1128/MCB.00256-14 (2014).
- 8 Adriaenssens, A. E. *et al.* Glucose-Dependent Insulinotropic Polypeptide Receptor-Expressing Cells in the Hypothalamus Regulate Food Intake. *Cell Metab* **30**, 987-996 e986, doi:10.1016/j.cmet.2019.07.013 (2019).
- 9 Kaneko, K. *et al.* Gut-derived GIP activates central Rap1 to impair neural leptin sensitivity during overnutrition. *J Clin Invest* **129**, 3786-3791, doi:10.1172/JCI126107 (2019).
- 10 Hauner, H., Glatting, G., Kaminska, D. & Pfeiffer, E. F. Effects of gastric inhibitory polypeptide on glucose and lipid metabolism of isolated rat adipocytes. *Annals of nutrition & metabolism* **32**, 282-288 (1988).
- 11 SAMMS, R. J. *et al.* 1009-P: The Dual GIP and GLP-1 Receptor Agonist Tirzepatide Regulates Lipid and Carbohydrate Metabolism through GIPR in Adipose Tissue. *Diabetes* **68**, 1009-P, doi:10.2337/db19-1009-P (2019).
- 12 Getty-Kaushik, L., Song, D. H., Boylan, M. O., Corkey, B. E. & Wolfe, M. M. Glucose-dependent insulinotropic polypeptide modulates adipocyte lipolysis and reesterification. *Obesity* **14**, 1124-1131, doi:10.1038/oby.2006.129 (2006).

- 13 Beck, B. & Max, J. P. Direct metabolic effects of gastric inhibitory polypeptide (GIP): dissociation at physiological levels of effects on insulin-stimulated fatty acid and glucose incorporation in rat adipose tissue. *Diabetologia* **29**, 68 (1986).
- 14 Beck, B. & Max, J. P. Hypersensitivity of adipose tissue to gastric inhibitory polypeptide action in the obese Zucker rat. *Cellular and molecular biology* **33**, 555-562 (1987).
- 15 Kim, S. J., Nian, C. & McIntosh, C. H. Resistin knockout mice exhibit impaired adipocyte glucose-dependent insulinotropic polypeptide receptor (GIPR) expression. *Diabetes* **62**, 471-477, doi:10.2337/db12-0257 (2013).
- 16 Storey, J. D. & Tibshirani, R. Statistical significance for genomewide studies. *Proc Natl Acad Sci U S A* **100**, 9440-9445, doi:10.1073/pnas.1530509100 (2003).
- 17 Wu, Q. *et al.* FATP1 is an insulin-sensitive fatty acid transporter involved in diet-induced obesity. *Molecular and cellular biology* **26**, 3455-3467, doi:10.1128/MCB.26.9.3455-3467.2006 (2006).

Reviewers' Comments:

Reviewer #1:

Remarks to the Author:

With this work, Killion et al. provide answers to the controversy of how both agonizing and antagonizing the GIP receptor are thought to be effective anti-obesity strategies. Using both in vivo models combined with a metabolomic assessment and a more detailed analysis on cellular levels, these well-designed studies are an important contribution to understanding the elusive GIP system. The manuscript is very well-written and the attention to detail that was applied when putting it together makes a reviewer's job very easy. Apart from a few minor remarks, I advise to accept this work for publication.

Minor remarks:

Figure 1A. This part lacks EC50 values, and could potentially be improved by choosing the same color code in Fig1a-c as in Fig1g-n

Line 66-72: When the authors compare antagonists with agonists, it would improve the understanding if they mention selectivity and PK for the antagonist

Line 299-311. Here the authors could consider to study arrestin recruitment under the same conditions and maybe also include subcellular localization of the ligand-receptor complex.

Reviewer #2:

Remarks to the Author:

The authors have adequately address this reviewer's comments.

Reviewer #3:

Remarks to the Author:

The authors have responded to the elements of my prior critique that are most important. Many of the responses are satisfactory. The authors make a convincing case that extended exposure of adipose cells to a GIPR agonist causes receptor internalization. The case that that receptor internalization leads to desensitization and equivalence between chronic agonism and antagonism is not as convincing.

1. The observation that the authors focus on explaining is the data in Fig 1g. Here obese mice lose weight whether treated for 3 wks with LA-GIP or the GIPRmAb. The implication of these results- that normal physiology includes a GIPR signal that contributes to maintenance of positive energy balance, is not explicitly discussed in this paper. Moreover, whether this physiologic signal is the result of circulating GIP or constitutive receptor activity is also not considered. So are the relative effects of the two treatments on GIPR activity- desensitization implies partial signaling while the antibody may completely block signaling; similarities and differences here could provide insight into the mechanism of action. In trying to rectify the common effects of chronic agonism and antagonism at a single receptor these issues need consideration.

2. While we are sympathetic to the authors explanation of differential effects of chronic GIPR agonism/antagonism on tissues that express the GIPR we do not find the experimental evidence convincing. The INS cells untreated with GIP have a nice cAMP dose-response to stimulation; the pretreated cells have a higher basal cAMP response (this does not appear to be a ceiling effect as the controls reach higher cAMP with stimulation) but do not respond to increasing doses of GIP. It is difficult not to see this as desensitization on some level. The authors suggest that low expression of the GIP receptor makes tissues more likely to be desensitized by chronic GIP stimulation. The test of this hypothesis in the data presented is not rigorous.

3. Urinary cAMP levels have been used in the past in clinical practice for diagnosing

hyperparathyroidism and have also been demonstrated to rise with exogenous glucagon treatment. We could find no evidence in the literature that plasma cAMP reflects the action of one of the many Gs coupled GPCR. However, the data in fig 2f appear unequivocally lower than controls and similar for chronic GIPR agonism and antagonism. What would make this data more convincing, and supportive of the authors core argument, would be to show that acute administration of DA-GIP increases circulating cAMP. If the argument in the paper is correct, the results should look much like the cort response in 2i.

4. Experiments 4m-p are important for the claim that GIPR signaling in adipose tissue contributes to body weight effects. The differences in the control and knockout mice is not large. For the authors claim to be valid the 2-way ANOVA should have a significant treatment x genotype interaction; this is not reported.

5. At some point in this line of investigation it may be useful to know what LA-GIP and GIPRmAb do to circulating GIP and leptin.

Reviewer 1

With this work, Killion et al. provide answers to the controversy of how both agonizing and antagonizing the GIP receptor are thought to be effective anti-obesity strategies. Using both in vivo models combined with a metabolomic assessment and a more detailed analysis on cellular levels, these well-designed studies are an important contribution to understanding the elusive GIP system. The manuscript is very well-written and the attention to detail that was applied when putting it together makes a reviewer's job very easy. A part from a few minor remarks, I advise to accept this work for publication.

Minor remarks:

Figure 1A. This part lacks EC50 values, and could potentially be improved by choosing the same color code in Fig1a-c as in Fig1g-n

We have now reported the EC50 values for Fig. 1A and changed the colors per your suggestion.

Line 66-72: When the authors compare antagonists with agonists, it would improve the understanding if they mention selectivity and PK for the antagonist

Thank you, we have now included the following statement:

"We previously reported that the muGIPR-Ab dose-dependently inhibited GIP-stimulated cAMP in vitro in the same assay reported here in Fig. 1A with $IC_{50} = 89.6$ nM, and in vivo, the maximum effect in the acute PD assay was achieved with muGIPR-Ab (25 mg/kg), which correlated with a mean serum concentration of 2250 nM, and allowed us to determine that muGIPR-Ab dosed 25 mg/kg every six days was sufficient to provide maximal target coverage¹."

Line 299-311. Here the authors could consider to study arrestin recruitment under the same conditions and maybe also include subcellular localization of the ligand-receptor complex.

We agree that this would be an interesting study to include, but unfortunately, we are not able to perform further lab work due to the ongoing COVID-19 pandemic. Since the addition of this data would not change any of this paper's conclusions, we have now added the following statement in the discussion to acknowledge the importance of these future studies:

"However, further work is needed to understand whether these cell type differences are due to differences in expression level or other components of cell signaling, including β -arrestin recruitment and subcellular localization of the ligand-receptor complex."

Reviewer 3

The authors have responded to the elements of my prior critique that are most important. Many of the responses are satisfactory. The authors make a convincing case that extended exposure of adipose cells to a GIPR agonist causes receptor internalization. The case that that receptor internalization leads to desensitization and equivalence between chronic agonism and antagonism is not as convincing.

1. The observation that the authors focus on explaining is the data in Fig 1g. Here obese mice lose weight whether treated for 3 wks with LA-GIP or the GIPRmAb. The implication of these results- that normal physiology includes a GIPR signal that contributes to maintenance of positive energy balance, is not explicitly discussed in this paper. Moreover, whether this physiologic signal is the result of circulating GIP or constitutive receptor activity is also not considered. So are the relative effects of the two treatments on GIPR activity- desensitization implies partial signaling while the antibody may completely block signaling; similarities and differences here could provide insight into the mechanism of action. In trying to rectify the common effects of chronic agonism and antagonism at a single receptor these issues need consideration.

We did not discuss these aspects as we have focused on the pharmacology of GIPR agonists and antagonists. It is well described in the literature that GIPR contributes to normal physiology as observed in mouse KO studies and we do not feel it needs repetition. The two treatments on GIPR activity are for the most part identical, as observed in all experiments we present with the single exception of IP-induced insulin secretion, and we have described this difference extensively. To address this reviewers' point, we have now added adipokines as another example of additional similarities that provide insight into the mechanism of action and will include these data in new Figure 1M-O. Notably, all adipokines measured were the same between the two treatments both alone and in combination with liraglutide as we have already seen reflected in body weight (Fig. 1G), food intake (Fig. 1H and Fig. 5N), fat mass (Fig. 1I), adipose tissue weights (Fig. 1J, K), liver weight (Fig. 1L), plasma metabolomics (Fig. 2A-H), and adipose tissue GIP-stimulated fatty acid uptake (Fig. 5U).

2. While we are sympathetic to the authors explanation of differential effects of chronic GIPR agonism/antagonism on tissues that express the GIPR we do not find the experimental evidence convincing. The INS cells untreated with GIP have a nice cAMP dose-response to stimulation; the pretreated cells have a higher basal cAMP response (this does not appear to be a ceiling effect as the controls reach higher cAMP with stimulation) but do not respond to increasing doses of GIP. It is difficult not to see this as desensitization on some level. The authors suggest that low expression of the GIP receptor makes tissues more likely to be desensitized by chronic GIP stimulation. The test of this hypothesis in the data presented is not rigorous.

We appreciate the reviewers' comment that the high basal activity with GIP pretreatment and lack of response to fresh GIP represents desensitization to some extent but the high basal levels, in contrast to the low levels in the other cell types, could reflect sustained activation, albeit not maximal. We have now added the statement that the high (submaximal) basal activity may be partially desensitized. We agree that our hypothesis that GIPR expression levels could determine if a cell type can become desensitized is not rigorously tested. We provide it as a hypothesis in our discussion and we have now toned down the language used in the discussion to make clear that it is simply our hypothesis and we have further emphasized that it is a hypothesis that requires future investigation.

Updated discussion paragraph:

"Conversely, in a pancreatic islet cell line, rat INS1 832/13, cells appear to not desensitize, but rather display high basal levels of cAMP after chronic GIPR agonism (Fig. 5H), suggesting that GIPR continues to signal in these cells. However, high basal activity with GIP pretreatment and lack of response to fresh

GIP represents desensitization to some extent but the high basal levels, in contrast to the low levels in the other cell types, could reflect sustained activation, albeit not maximal. Similarly, mice treated with LA-Agonist for 21 days were able to respond to DA-GIP stimulated insulin secretion 4 hours after the last LA-Agonist dose (Fig. 1P), which suggests that GIPR in pancreatic β -cells does not desensitize in a manner observed adipocytes or neuronal cells. We hypothesize that cell type differences in desensitization can be explained by differences in GIPR expression where pancreatic islets have dramatically higher levels of *Gipr* expression compared to adipose tissue and brain sections (Supplemental Fig. 4). However, this is a hypothesis that requires further work to understand whether these cell type differences are due to differences in expression level or other components of cell signaling, including β -arrestin recruitment and subcellular localization of the ligand-receptor complex.

3. Urinary cAMP levels have been used in the past in clinical practice for diagnosing hyperparathyroidism and have also been demonstrated to rise with exogenous glucagon treatment. We could find no evidence in the literature that plasma cAMP reflects the action of one of the many Gs coupled GPCR. However, the data in fig 2f appear unequivocally lower than controls and similar for chronic GIPR agonism and antagonism. What would make this data more convincing, and supportive of the authors core argument, would be to show that acute administration of DA-GIP increases circulating cAMP. If the argument in the paper is correct, the results should look much like the cort response in 2i.

While we agree that cAMP levels in plasma is unusual to reflect the action of GPCRs, it was still a detected metabolite in our unbiased metabolomics study and supports our contention that agonism mimics antagonism. While we agree the figure could be strengthened by adding the stimulation of cAMP by GIP, we are currently unable to conduct in vivo experiments due to the ongoing COVID-19 pandemic. We do not feel the additional experiment change any of the conclusions in the manuscript as the key data is already included, that is, agonism or antagonism leads to the same response, lower cAMP levels, and this is consistent with our prevailing hypothesis, which is supported on many levels. We have now added the reviewer's point to the manuscript text:

"It is important to note that there is no evidence in the literature that plasma cAMP reflects the action of a single Gs coupled GPCR, but the plasma cAMP data in Fig. 2F is unequivocally lower than controls and similar for chronic GIPR agonism and antagonism."

4. Experiments 4m-p are important for the claim that GIPR signaling in adipose tissue contributes to body weight effects. The differences in the control and knockout mice is not large. For the authors claim to be valid the 2-way ANOVA should have a significant treatment x genotype interaction; this is not reported.

Yes, agreed. Table 1 is the ANOVA table for Fig. 4m-n and Table 2 is the ANOVA table for Fig. 4o-p demonstrating significant treatment x genotype interactions with $p < 0.0001$, and we have now included this in the manuscript text.

2way ANOVA Tabular results				
Table Analyzed	Body Weight % Change_Dosing_Males			
Two-way RM ANOVA	Matching: Stacked			
Alpha	0.05			
Source of Variation	% of total variation	P value	P value summary	Significant?
Interaction	8.542	<0.0001	****	Yes
Time	10.58	<0.0001	****	Yes
Column Factor	26.98	0.0012	**	Yes
Subjects (matching)	36.31	<0.0001	****	Yes

Table 1 (for data in Fig. 4m-n):

Table 2 (for data in Fig. 4o-p):

2way ANOVA Tabular results				
Table Analyzed	Body Weight % Change_Dosing_Females			
Two-way RM ANOVA	Matching: Stacked			
Alpha	0.05			
Source of Variation	% of total variation	P value	P value summary	Significant?
Interaction	5.88	<0.0001	****	Yes
Time	37.98	<0.0001	****	Yes
Column Factor	21.88	0.0196	*	Yes
Subjects (matching)	24.46	<0.0001	****	Yes

5. At some point in this line of investigation it may be useful to know what LA-GIP and GIPRmAb do to circulating GIP and leptin.

We have previously reported GIP levels in our previous publication where muGIPR-Ab does not alter plasma GIP levels¹ (reference Supplemental Fig. 3E) and we have now included plasma leptin here in Fig. 1M where both muGIPR-Ab and LA-Agonist significantly reduced leptin both alone and in combination with liraglutide in a similar manner for both treatments. We are unable to measure endogenous GIP in the LA-Agonist treated animals because of the cross-reactivity with the LA-Agonist peptide; however, we now include leptin levels and also other measured adipokines (new Fig. 1M-O). Notably, other companies working on GIPR agonists have not reported endogenous GIP levels after GIPR agonist treatment likely owing to the same issue with cross-reactivity.

Reviewers' Comments:

Reviewer #3:

Remarks to the Author:

No new comments

David D'Alessio